# HSDF: Hybrid Sign and Distance Field
# for Modeling Surfaces with Arbitrary Topologies

**Li Wang**[1,2], **Jie Yang**[1,2], **Weikai Chen**[3],
**Xiaoxu Meng**[3], **Bo Yang**[3], **Jintao Li**[1,2], **and Lin Gao** (✉)[‡1,2]

[1]Beijing Key Laboratory of Mobile Computing and Pervasive Device, Institute of Computing
Technology, Chinese Academy of Sciences
[2]University of Chinese Academy of Sciences
[3]Tencent Games Digital Content Technology Center
{wangli20s, yangjie01}@ict.ac.cn    chenwk891@gmail.com
{xiaoxumeng,brandonyang}@tencent.com    {jtli, gaolin}@ict.ac.cn

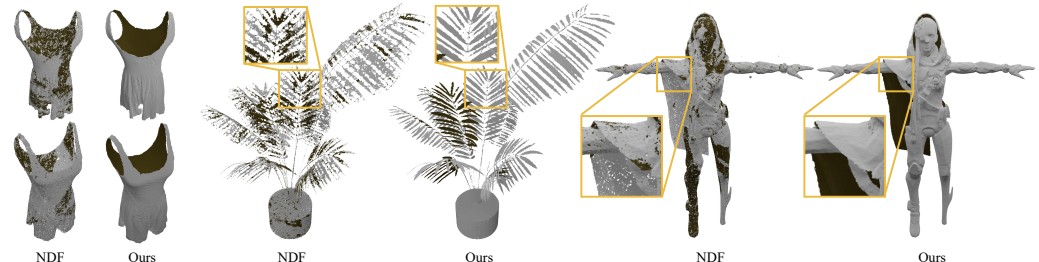

Figure 1: We reconstruct three groups of representative objects with both open and closed surfaces using NDF [15] (left) and our proposed HSDF (right). Compared with the SOTA NDF [15], our method achieves higher surface reconstruction quality and more consistent surface normals. All the results are reconstructed with an equivalent resolution.

## Abstract

Neural implicit function based on signed distance field (SDF) has achieved impressive progress in reconstructing 3D models with high fidelity. However, such approaches can only represent closed surfaces. Recent works based on unsigned distance function (UDF) are proposed to handle both watertight and open surfaces. Nonetheless, as UDF is signless, its direct output is limited to the point cloud, which imposes an additional challenge on extracting high-quality meshes from discrete points. To address this challenge, we present a novel neural implicit representation coded HSDF, which is a hybrid of signed and unsigned distance fields. In particular, HSDF is able to represent arbitrary topologies containing both closed and open surfaces while being compatible with existing iso-surface extraction techniques for easy field-to-mesh conversion. In addition to predicting a UDF, we propose to learn an additional sign field. Unlike traditional SDF, HSDF is able to locate the surface of interest before level surface extraction by generating surface points following NDF [15]. We are then able to obtain open surfaces via an adaptive meshing approach that only instantiates regions containing surfaces into a polygon mesh. We also propose HSDF-Net, a dedicated learning framework that factorizes the learning of HSDF into two easier sub-problems. Experiments and evaluations show that HSDF outperforms the state-of-the-art techniques both qualitatively and quantitatively.

---

[*]Corresponding author is Lin Gao (gaolin@ict.ac.cn).

36th Conference on Neural Information Processing Systems (NeurIPS 2022).

# 1 Introduction

Recent advances in neural implicit representation [12, 40, 43, 48, 58] have set a new state of the art in 3D modeling and reconstruction by breaking the previous barrier in resolution and topology. However, as such approaches rely on the signed distance function (SDF) which divides the space into inside and outside of the object, they are limited to representing closed shapes. To lift the limitation, methods based on unsigned distance function (UDF) [15, 52, 50] are proposed such that a much broader class of shapes containing open surfaces can be effectively represented and learned via deep neural networks.

However, as UDF is signless, directly applying the iso-surface extracting technique, e.g. the Marching Cubes algorithm, would convert all open surfaces into the closed mesh. To generate open structures, these approaches have to convert the resulting UDF field into discrete points and then apply the Ball-Pivoting algorithm [4] (BPA) to obtain the meshing result. Nonetheless, the BPA technique is prone to introduce self-intersections and disconnected surface patches with inconsistent normals (see Figure 1). In addition, BPA is highly sensitive to the input parameters and often requires per-shape parameter tuning in order to generate a complete meshing result. This hinders UDF-based approaches from being practically used in real-world applications as mesh remains the prominent standard for modeling and rendering in both industry and academia.

To address the above issue, we present a novel learnable implicit representation, named *Hybrid Sign and Distance Function (HSDF)*, that can faithfully represent complex geometry containing both closed and open surfaces, while being compatible with off-the-shelf iso-surface extraction methods, e.g. the Marching Cubes algorithm, for easy and high-quality field-to-mesh conversion. The key idea of HSDF is to integrate the advantages of both SDF and UDF while avoiding their respective shortcomings. We empirically find that the learning of UDF is quite robust and can generalize well to novel data. Therefore, to inherit the benefit of UDF and overcome its limitation, we propose to learn an additional sign field in addition to UDF via a sign predictor. Unlike traditional SDF, HSDF is able to locate the open surface before performing level surface extraction. We achieve this by generating surface points via the gradient field of the unsigned distance function following NDF [15]. Hence, we are able to create local SDFs by multiplying the UDF with the sign field and cast complex shapes containing both closed and open surfaces by incorporating an adaptive meshing algorithm that only instantiates the regions containing surface points into a polygon mesh.

As HSDF may be discontinuous at regions far from the surface due to the sudden change of signs, we propose to factorize the learning of HSDF into two easier sub-tasks, each of which learns a continuous function. Specially, we propose HSDF-Net that learns the unsigned distance and sign field individually using the same input in a separate manner. A fusion framework is also proposed to faithfully fuse the predicted distance and sign field at the inference time. Finally, we introduce an adaptive masked Marching Cubes algorithm for converting HSDF into the high-quality mesh at flexible resolution efficiently. Extensive experiments show that HSDF can outperform the previous state-of-the-art methods in both qualitative and quantitative measurements. We summarize the key contributions as follows:

- We introduce HSDF, a novel neural implicit field that can faithfully represent complex shapes with closed and open surfaces while being compatible with existing level surface extraction techniques.

- We propose a dedicated learning framework that separates the decoding of sign and distance in accordance with the definition of HSDF.

- We propose an adaptive masked Marching Cubes algorithm to generate meshes from HSDF efficiently.

# 2 Related Works

3D neural shape representation is a very fundamental and popular research topic in computer vision and graphics. In this section, we will review some recent advances in neural shape representations from two aspects: Explicit & Implicit Representation. Some survey papers [30, 6, 1, 60] have comprehensively summarized the development of 3D shape representation.

**Learning on 3D Explicit Representations.** For learning on 3D shapes, various 3D representations are applied to the learning-based approaches, such as point clouds, voxels, and polygonal meshes. The voxel represents the concrete 3D shapes with regular domain which is directly an extension of 2D pixel grids, and it is widely used in 3D deep learning. Many previous works [38, 18, 59] investigated this representation by extending the 2D deep learning operators to the 3D domain. However, 3D voxel representation suffers from low-resolution, high-computation, and memory-consumption drawbacks. Further, Wang et al.tried their best to address the above drawbacks by introducing the octree-based deep learning methods [54, 56, 55, 36]. The point cloud is flexible and easier to be captured by some portable 3D scanners (*e.g.* Microsoft Kinect), but the main challenge is that the fidelity is too low to represent high-quality shape geometry and topology. Due to its irregular and disordered structure, the traditional deep learning methods cannot directly be applied, Qi et al.firstly introduced PointNet [45] and PointNet++ [46] for 3D classification and segmentation by pooling operation which is order-independent. Wang et al. [57] constructed a sub-graph by KNN and applied the graph neural network on the point cloud to learn the local geometric features for shape analysis [8]. There is also work [29] that leverages hierarchical structure for point cloud understanding. Some works [11, 34, 26, 47, 35, 28] exploited the topology of polygonal meshes for shape analysis. There is a recent trend [24, 53, 31, 37, 42, 22, 62] of studying the deformable meshes to represent the detailed geometry. The predefined connections of the vertices set to limit the flexibility and topology.

**Learning on 3D Implicit Representations.** Implicit function is a powerful tool capable of modeling arbitrary geometric details and topological structures. With the help of deep neural networks, a complex shape can be implicitly represented by classifying the query point in/outside a shape (Binary Occupancy) [39, 44, 17, 23, 49, 20, 13] or predicting the continuous signed distance (SDF) to the shape surface [12, 40, 43, 32, 48], where the sign of SDF determines the query point in/out-side of a shape. The binary occupancy and signed distance function can only represent the watertight closed surface, since it is essential that partition the set of query points inside or outside of the shape. For the open surfaces (*e.g.* cloth, thin-shell models), Atzmon et al. [2] developed a deep learning approach – Sign Agnostic Learning (SAL) for raw data with any inside/outside labeling, but the prediction of SAL is still SDF for the close surface reconstruction. Furthermore, the concurrent works NDF [16] and DUDE [52] proposed neural-based methods that predict the *unsigned* distance field to represent arbitrary surfaces without any closed surface data. NDF adopts the multi-scale encoding techniques from IF-Net [14] to enhance the surface details. DUDE introduces a disentangled shape representation that utilizes the unsigned distance field (UDF) and normal vector field (nVF), which learns high-fidelity representations. Due to the gradient vanishing of UDF close to the surface, UDF fails to predict high-quality estimation of high-ordered geometry attributes (*e.g.* normal, tangent plane), CSPNet [51] represents such surfaces by a class of implicit representation – closest-surface-point (CSP), which achieves local geometric attributes accurately and efficiently.

Recent concurrent works such as GIFS [63], DeepCurrents [41], MeshUDF [25], 3PSDF [9] and NDC [10] also focus on modeling open surfaces. DeepCurrents [41] combines the explicit boundary curves and the implicit field. And the final open surfaces are obtained by solving a minimal surface problem, which is not as flexible as pure implicit field to model arbitrary shapes. Methods like GIFS [63] and NDC [10] model the relationships between every two points or edge intersections, which is not as efficient as modeling the relationships between points and surfaces. And the reconstructed surface normals will be different from the ground truth meshes because point-surface relationships are missing. MeshUDF [25] proposes a differential meshing algorithm for UDF. Since its meshing process is still based on the Marching Cubes algorithm and the signs required are decided in a heuristic way, the outcome meshes suffer from flipped faces and have different normal directions from ground truth meshes. 3PSDF [9] proposes an implicit representation that divides the space into three classes, namely positive, negative and null. Null class enables 3PSDF to mask out certain regions in the space and represent open shapes. 3PSDF formulates the reconstruction task as a classification problem. It eases the learning difficulty but also limits the applications on downstream tasks that require SDF continuity, such as neural rendering.

In contrast, HSDF can efficiently model arbitrary shapes with both open and closed surfaces and achieves high-quality reconstruction with more consistent surface normals similar to the ground truth using the proposed adaptive masked Marching Cubes algorithm.

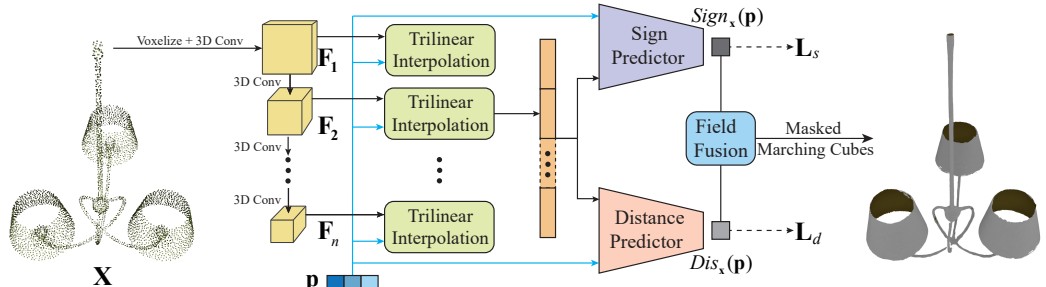

Figure 2: HSDF-Net architecture. The input sparse point cloud is voxelized and encoded in a multi-scale manner into a shape code. Next, the distance predictor takes the shape code and query point $\mathbf{p}$ as input to predict an unsigned distance $Dis(\mathbf{p})$. The sign predictor takes the same input and predicts a signed value $Sign(\mathbf{p})$. The field fusion module (Sec. 3.3) is proposed to fuse $Dis(\mathbf{p})$ and $Sign(\mathbf{p})$. The rightmost lamp example is reconstructed from test data using the proposed adaptive masked Marching Cubes algorithm (Sec. 3.4).

## 3 Method

In this section, we introduce the proposed shape representation and our HSDF-Net with an adaptive masked Marching Cubes algorithm into high-fidelity single-layer meshes. A diagram of our system architecture is shown in Fig. 2. We first define the proposed implicit shape representation (Sec. 3.1). Then we introduce the model structure of HSDF-Net and explain our learning strategy (Sec. 3.2). Finally, we illustrate the approach to fuse the hybrid distance and sign together to form a HSDF (Sec. 3.3) and the adaptive masked Marching Cubes algorithm (Sec. 3.4).

### 3.1 Shape Representation

We define the target surface as $\mathcal{S}$. Our proposed formulation takes a vectorized shape code $\mathbf{z} \in \mathcal{Z}$ and a point $\mathbf{p} \in \mathbb{R}^3$ as input, and predicts a hybrid sign and distance function $\mathrm{HSDF}(\mathbf{p}, \mathbf{z}) : \mathbb{R}^3 \times \mathcal{Z} \mapsto \mathbb{R}$. If we define $\mathbf{p}'$ as the closest point of $\mathbf{p}$ on the surface $\mathcal{S}$ and the surface normal of $\mathbf{p}'$ as $\mathbf{v}$. Then unsigned distance can be computed as $\mathrm{UDF}(\mathbf{p}, \mathbf{z}) = \|\mathbf{p} - \mathbf{p}'\|_2$ and the sign is calculated as $\mathrm{SIGN}(\mathbf{p}, \mathbf{z}) = \mathrm{sign}(\mathbf{v} \cdot (\mathbf{p} - \mathbf{p}'))$, where $\mathrm{sign}(\cdot)$ returns the sign of the input. Specifically, $\mathrm{SIGN}(\mathbf{p}, \mathbf{z})$ is positive if $\mathbf{v}$ aligns with $\mathbf{p} - \mathbf{p}'$ and negative otherwise. The overall HSDF expression can be defined as:

$$
\begin{aligned}
\mathrm{HSDF}(\mathbf{p}, \mathbf{z}) &= \mathrm{SIGN}(\mathbf{p}, \mathbf{z}) \cdot \mathrm{UDF}(\mathbf{p}, \mathbf{z}) \\
&= \mathrm{sign}(\mathbf{v} \cdot (\mathbf{p} - \mathbf{p}')) \cdot \|\mathbf{p} - \mathbf{p}'\|_2 \\
\text{s.t. } \mathbf{p}' &= \operatorname*{argmin}_{\mathbf{p}_s \in \mathcal{S}} \|\mathbf{p} - \mathbf{p}_s\|_2
\end{aligned}
\tag{1}
$$

By this definition, we are able to locate the surface of interest by searching for the points on the surface following the paradigm of [15] on leveraging the gradient of $\mathrm{UDF}(\cdot)$. In addition, the introduced sign field SIGN can convert the UDF into a signed one, enabling the use of the iso-surface extraction technique for easy field-to-mesh conversion. By incorporating an adaptive masked Marching Cubes algorithm (detailed in Section 3.4) that only instantiates the regions containing surface points into a mesh, we are able to model complex topology with open surfaces.

### 3.2 HSDF-Net: Model Architecture

For regions that are far from the surface, the HSDF representation may be discontinuous as the sign field may undergo a sudden change. To ease the learning of HSDF, we factorize the learning into two sub-problems: learning of the unsigned distance and the sign field. As depicted in Fig. 2, we propose HSDF-Net that contains three components: a shared volume encoder, a distance predictor, and a sign predictor. The shape encoder extracts features from input sparse point clouds in a multi-scale manner, which follows IF-Net [14]. Then we feed encoded multi-scale shape code and query point $\mathbf{p}$ into the sign and the distance predictors in Fig. 2 to get a sign prediction $Sign(\mathbf{p})$ and a distance prediction $Dis(\mathbf{p})$. Next, we will introduce our network components' formulation in detail.

**Volume Encoder:** Our goal is to reconstruct an arbitrary surface $\mathcal{S}$ from a sparse point cloud $\mathbf{X} \in \mathcal{X}$. The input point cloud is first voxelized and encoded by 3D CNNs into multi-scale grid features $\mathbf{F}_1, \cdots, \mathbf{F}_n, \mathbf{F}_k \in \mathcal{F}_k^{K \times K \times K}$, where $K$ is the grid size and $\mathcal{F}_k \in \mathbb{R}^C$ is a feature with C channels. If we denote the result of trilinear interpolation of grid feature $\mathbf{F}_i$ at position $\mathbf{p}$ as $\mathbf{T}_i$, then $\Psi_{\mathbf{x}}(\mathbf{p}) = (\mathbf{T}_1, \ldots, \mathbf{T}_n)$ can represent the concatenated shape code for input $\mathbf{X}$ at point $\mathbf{p}$ and $\Psi_{\mathbf{x}}(\mathbf{p}) : \mathbb{R}^3 \mapsto \mathcal{F}_1 \times \ldots \times \mathcal{F}_n$ defines the encoder function, see IF-Net [14].

**Distance Predictor:** The distance predictor predicts the UDF between points and the ground truth surface. We formulate this as a function $\Phi((\mathbf{T}_1, \ldots, \mathbf{T}_n)) : \mathcal{F}_1 \times \ldots \times \mathcal{F}_n \mapsto \mathbb{R}^+$, and we use fully connected layers with ReLU activation functions to guarantee $\Phi \geq 0$. In general, the distance predictor could be formulated as $Dis_{\mathbf{x}}(\mathbf{p}) = (\Phi \circ \Psi_{\mathbf{x}})(\mathbf{p}) : \mathbb{R}^3 \mapsto \mathbb{R}_0^+$.

**Sign Predictor:** We extend sign concept to arbitrary shapes. For example, a T-shirt model has consistent outward face normal directions, and we can easily tell the sign of a query point $\mathbf{p}$ even though a T-shirt is obviously an open surface because we will intuitively choose the closest point $\mathbf{p}'$ on the surface and use the normal direction of $p'$ as a clue of the sign. Detailed formulation is defined above in Sec. 3.1. For cars with inner structures, we also compute the sign of a query point $\mathbf{p}$ according to the normal direction of its closest surface point, no matter whether $\mathbf{p}$ is inside the car or close to the outside surface.

Therefore, we extend the concept of "signed distance" to a broader range of shapes like open shapes by combining "distance" and "sign" computation and prediction processes. We define

(a) Sign Function    (b) Distance Function    (c) Original HSDF

(d) Sign Function    (e) Distance Function    (f) Optimized HSDF

- positive region
- negative region
- zero level-set surface
- sign function gradient
- distance function gradient

Optimized Point A:

Optimized Point B:

Figure 3: 2D illustration of sign and distance fusion. Assume the green line is the target surface. **Row 1:** By multiplying sign and distance pointwise, we can obtain an original HSDF, where some points (e.g. A and B) may be wrongly signed. **Row 2:** By using gradients of sign function (*i.e.* yellow vectors) and gradients of distance function (*i.e.* brown vectors) to optimize the original fused HSDF, HSDF of points A, B can be effectively rectified (*i.e.* green line).

$\Theta((\mathbf{T}_1, \ldots, \mathbf{T}_n)) : \mathcal{F}_1 \times \ldots \times \mathcal{F}_n \mapsto \mathbb{R}_0$ as a sign predictor. By composition we obtain our sign predictor $Sign_{\mathbf{x}}(\mathbf{p}) = (\Theta \circ \Psi_{\mathbf{x}})(\mathbf{p}) : \mathbb{R}^3 \mapsto \mathbb{R}_0$ which we treat as a regressor to the signed distance. Hence, the predicted binary sign can be computed as $sign(Sign_{\mathbf{x}}(\mathbf{p})) : \mathbb{R}^3 \mapsto [-1, 1]$ where $\text{sign}(\cdot)$ returns the sign of the input. And the ground truth binary sign can also be represented as $SIGN_{\mathbf{x}}(\mathbf{p}) = sign(\text{SDF}(\mathbf{p}, \mathcal{S}_{\mathbf{x}})) : \mathbb{R}^3 \mapsto [-1, 1]$.

**Learning:** We define the volume encoder, distance predictor and sign predictor as $\mathbf{w_e}$, $\mathbf{w_d}$ and $\mathbf{w_s}$, respectively. During distance regression training, we use $L_1$ mini-batch loss to jointly optimize $\mathbf{w_e} \cup \mathbf{w_d}$:

$$\mathcal{L}_d := \sum_{\mathbf{x} \in \mathcal{B}} \sum_{\mathbf{p} \in \mathcal{P}} |\min(Dis_{\mathbf{x}}(\mathbf{p}), \delta) - \min(\text{UDF}(\mathbf{p}, \mathcal{S}_{\mathbf{x}}), \delta)|$$

where $\mathcal{B}$ and $\mathcal{P}$ stand for a mini-batch and a sub-sample of points. Clamping the maximal regressed distance to value $\delta > 0$ forces the model capacity to focus more on the vicinity of the surface.

During sign predictor training, we use $L_1$ mini-batch loss to jointly optimize $\mathbf{w_e} \cup \mathbf{w_s}$:

$$\mathcal{L}_s := \sum_{\mathbf{x} \in \mathcal{B}} \sum_{\mathbf{p} \in \mathcal{P}} |\max(\min(Sign_{\mathbf{x}}(\mathbf{p}), \delta), -\delta) - \max(\min(\text{SDF}(\mathbf{p}, \mathcal{S}_{\mathbf{x}}), \delta), -\delta)|$$

Finally, we train our HSDF-Net in an end-to-end manner by optimizing the total loss: $\mathcal{L}_{total} := \mathcal{L}_d + \mathcal{L}_s$.

## 3.3 Sign and Distance Field Fusion

The HSDF-Net trains a shared volume encoder and two decoders respectively for the sign and distance prediction. With this learning manner, we can obtain a predicted sign function $Sign_{\mathbf{x}}(\mathbf{p})$ and a predicted distance function $Dis_{\mathbf{x}}(\mathbf{p})$ of query points $\mathbf{p}$ after training. Although $Sign_{\mathbf{x}}(\mathbf{p})$ and

$Dis_{\mathbf{x}}(\mathbf{p})$ are both learned continuous function, they can be fused into a hybrid sign and distance function HSDF($\mathbf{p}, \mathcal{S}$) with our fusing algorithm illustrated in Fig. 3. In Fig. 3, we plot a local region in 2D of learned sign function $Sign_{\mathbf{x}}(\mathbf{p})$ and distance function $Dis_{\mathbf{x}}(\mathbf{p})$ where $\mathbf{X}$ and $\mathbf{p}$ is the input sparse point cloud and the query point, respectively. The first row of Fig. 3 describes a straightforward fusion strategy that multiply distance with sign pointwisely: HSDF($\mathbf{p}, \mathbf{X}$) = $Sign_{\mathbf{x}}(\mathbf{p}) \cdot Dis_{\mathbf{x}}(\mathbf{p})$.

In the second row of Fig. 3, we design a simple but effective optimization using the gradients of the predicted sign function $Grad_s = \nabla Sign_{\mathbf{x}}(\mathbf{p})$ and the gradients of the predicted distance function $Grad_d = \nabla Dis_{\mathbf{x}}(\mathbf{p})$ to optimize the predicted signs of points like A, B. As described in Fig. 3, if $Grad_s \cdot Grad_d$ is negative as point A, then the sign will be optimized to negative and vice versa for optimizing point B. Since $Grad_s$ and $Grad_d$ can be inferred directly using the backward operation of deep neural networks, this optimization is not time-consuming.

### 3.4  Mesh Extraction

Our fused HSDF enables us to use efficient meshing methods like Marching Cubes. Hence, we propose an adaptive masked Marching Cubes algorithm. A 2D T-shirt example of fused HSDF is depicted on the left side in Fig. 4. Assume the green contour stands for the open surface, which disconnects at the collar, sleeves, and waist. The red/ blue region indicates where the sign is positive/ negative.

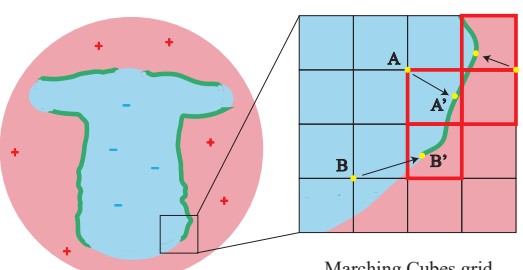

Marching Cubes grid

Figure 4: A 2D illustration of mesh extraction on our fused HSDF. Green contour indicates an open shape in 2D. We push Marching Cubes grid points like $A$ and $B$ to their closest points on the surface, i.e., $A'$ and $B'$, respectively, using the gradients of distance function (*i.e.* black vectors). All the boxes in red enclosing surfaces form a mask for Marching Cubes to extract the complex shapes into meshes.

Next, we set up a Marching Cubes grid to mesh the surface $\mathcal{S}$ in green. All the grid points can be pushed to the underlying surface by the gradients of the distance function. Among them, we take points like A, B for example. We define $\mathbf{p}_A$ and $\mathbf{p}_B$ as the coordinate of A and B. Then the coordinate of the closest surface point of A (*i.e.* $A'$) can be computed as $\mathbf{p}_{A'} = \mathbf{p}_A - Dis_{\mathbf{x}}(\mathbf{p}_A) \cdot \frac{\nabla Dis_{\mathbf{x}}(\mathbf{p}_A)}{||\nabla Dis_{\mathbf{x}}(\mathbf{p}_A)||_2}$ where $-\frac{\nabla Dis_{\mathbf{x}}(\mathbf{p}_A)}{||\nabla Dis_{\mathbf{x}}(\mathbf{p}_A)||_2}$ represent a unit vector towards the underlying surface. And we compute $\mathbf{p}_{B'}$ and all other points by the same process. After all the grid points are pushed to the surface, we can easily pick out those boxes (in red) which enclose surfaces. And these selected cubes form a mask for us to find out where the surfaces lie (works for arbitrary shapes). Finally, simple masked Marching Cubes can be applied to our HSDF to extract any complex shapes.

## 4  Experiments

In this section, we validate HSDF on 3D shape reconstruction from sparse point clouds and demonstrate that HSDF can reconstruct watertight shapes on par with SOTAs. And then we show that our approach outperforms SOTA techniques on reconstructing complex shapes such as cars with inner structures in terms of quality and speed.

### 4.1  Data Preparation and Metrics

**HSDF Computation.** The HSDF computation follows a similar manner as mesh-to-sdf [33]. For every spatial point $\mathbf{p}$, we find the nearest point $\mathbf{p}'$ on the mesh surface and its corresponding surface normal $\mathbf{v}$. If the dot product of $\mathbf{v}$ and $\mathbf{p}' - \mathbf{p}$ is negative, we assign its sign to be positive with its distance to be $|\mathbf{p}' - \mathbf{p}|$ and vice versa. Please refer to the supplementary for further experimental details and results.

**Computation for Complex Shapes.** We compute HSDFs on repaired ShapeNet dataset [19] on four typical categories with complex shapes, namely cars (7,497 models) with inner structures, chairs (6,579 models), lamps (2,316 models), and ships (1,851 models) with thin open surfaces. We also

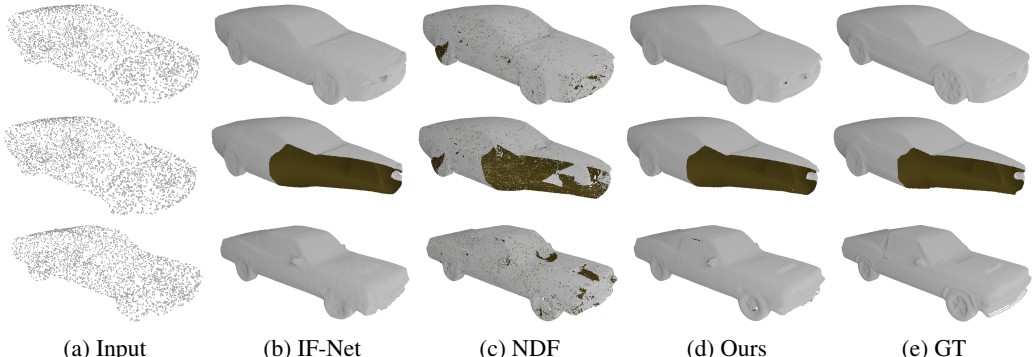

|    (a) Input    |    (b) IF-Net    |    (c) NDF    |    (d) Ours    |    (e) GT    |

Figure 5: Comparison between different methods on reconstructing closed shapes (without inner structure) from point clouds. The first row and the third row are independent experiments. The second row visualizes the inner reconstruction of the example in the first row. Input point number is $3,000$ for all the results.

attain HSDFs on 307 garments from MGN [5] because garments are also a typical category comprised of open surfaces.

**Computation for Closed Shapes.** In order to compare the performance on watertight shapes, we compute HSDFs on 3,094 ShapeNet [7] cars, which have been converted to closed shapes by DISN [61]. If the model only contains closed surfaces, HSDF is reduced to vanilla SDF, which also conforms to our intuition, because HSDFs are the natural extension of SDFs to represent a wider range of shapes.

**Metrics.** We follow the commonly used reconstruction metrics to evaluate the reconstructed meshes' quality: Chamfer distance (CD) [3], Normal consistency (NC) [39] and Oriented Normal Consistency which takes face directions into account. OccNet [39] defines a normal consistency score as the mean **absolute** dot product of the normals in one mesh and the normals at the corresponding nearest neighbors in the other mesh. In contrast, we define Oriented Normal Consistency as the mean dot product of the normals in one mesh and the normals at the corresponding nearest neighbors in the other mesh to make it sensitive to flipped mesh faces. More implementation details are demonstrated in supplemental materials, and We will release the code and data for facilitating future research.

## 4.2 Shape Fitting of Single Model

To demonstrate HSDFs can represent complex shapes with both closed and open surfaces, and can be learned by HSDF-Net, we fit models using NDF [15] and our HSDF-Net. For NDF, we extract meshes following the same process as NDF [15], namely using the Ball-Pivoting algorithm to mesh dense point clouds generated by a trained NDF.

As shown in Fig. 1, NDF results in poor quality. The holes and flipped faces cannot be easily optimized by post-processing techniques like closing holes or recomputing normals. Our approach can robustly reconstruct complex shapes with both open and closed surfaces as shown in Fig. 1, such as the dresses, plant leaves, and human characters. We also compare the timing and storage consumption between two single-layer mesh prediction approaches: HSDF-Net with MCubes (ours) and NDF. As shown in Table 1, our approach outperforms NDF in time and storage budget.

| Cost | Method | \multicolumn{3}{c}{Resolution} | | |
| | | $64^3$ | $128^3$ | $256^3$ |
|---|---|---|---|---|
| Time | NDF | 89s | 58m | 780m |
| | Ours | **5s** | **17s** | **95s** |
| Storage | NDF | 17M | 64M | 1276M |
| | Ours | **1M** | **3M** | **10M** |

Table 1: Comparison on average meshing time and storage budget per model. Our HSDF can reconstruct single-layer meshes much faster with a much lower storage budget.

## 4.3 Shape Reconstruction of Closed Surfaces

| Dataset | | MGN | | Car | | Chair | | Ship | | Lamp | | Mean | |
|---|---|---|---|---|---|---|---|---|---|---|---|---|---|
| #Points | | 10K | 3K | 10K | 3K | 10K | 3K | 10K | 3K | 10K | 3K | 10K | 3K |
| Chamfer Distance | Input | 0.879 | 1.70 | 0.683 | 1.91 | 0.814 | 2.42 | 0.327 | 0.933 | 0.350 | 1.08 | 0.611 | 1.61 |
| | W.GT | 3.46 | 3.46 | 4.87 | 4.87 | 2.93 | 2.93 | 3.79 | 3.79 | 5.32 | 5.32 | 4.07 | 4.07 |
| | NDF | **0.102** | 0.158 | 0.137 | 0.325 | 0.122 | 0.291 | **0.104** | 0.207 | **0.142** | **0.231** | **0.121** | 0.242 |
| | Ours | 0.114 | **0.151** | **0.124** | **0.289** | **0.101** | **0.209** | 0.133 | **0.194** | 0.166 | 0.274 | 0.128 | **0.223** |
| Normal Consistency | NDF | 0.954 | 0.938 | 0.879 | 0.817 | 0.903 | 0.865 | 0.884 | 0.850 | 0.909 | 0.869 | 0.906 | 0.868 |
| | Ours | **0.962** | **0.957** | **0.898** | **0.842** | **0.920** | **0.896** | **0.891** | **0.863** | **0.923** | **0.904** | **0.919** | **0.892** |
| Oriented NC | NDF | 0.517 | 0.390 | 0.383 | 0.310 | 0.423 | 0.344 | 0.445 | 0.358 | 0.541 | 0.505 | 0.462 | 0.381 |
| | Ours | **0.929** | **0.900** | **0.711** | **0.646** | **0.835** | **0.749** | **0.826** | **0.732** | **0.878** | **0.760** | **0.836** | **0.757** |

Table 2: Comparison on Chamfer distance($\times 10^{-4}$) & Normal Consistency & Oriented Normal Consistency which is sensitive to face normal directions.

Most of the SOTA such as OccNet [39] and IF-Net [14] require watertight training data. In order to compare with these SOTA methods, we train these methods along with NDF [15] on 3094 ShapeNet [7] cars pre-processed by DISN [61]. The pre-processing transform open shapes into closed shapes and removes all the

| # Points | Input | OccNet | IF-Net | NDF | Ours |
|---|---|---|---|---|---|
| 300 | 12.5 | 2.53 | 2.76 | 1.51 | **1.34** |
| 3000 | 1.54 | 1.17 | 0.615 | 0.191 | **0.155** |

Table 3: Comparison on Watertight Cars (DISN dataset) using Chamfer-$L_2$ Distance($\times 10^{-4}$).

interior contents of cars like seats. In Table 3, we compare the Chamfer Distance [21] of our method against all the baselines quantitatively with 300 input points and 3,000 input points. Our method achieves the best performance. In Fig. 5, we show that our HSDFs can achieve comparable quality as UDF-based SOTA [15] and SDF-based SOTA [14]. All the results are reported using test data unseen during training time.

## 4.4 Shape Reconstruction of Complex Shapes

In order to demonstrate that HSDFs can reconstruct meshes with better quality efficiently, we train our HSDF-Net on five representative categories containing complex shapes illustrated in Sec. 4.1. In Table 2, we compare our method quantitatively against NDF and watertight ground truth (W.GT) on the complex shapes with 3,000 and 10,000 points as input. The visual comparisons are shown in Fig. 6 where we can see thin open structures (*i.e.* the clothes, the back of the chair, and the hat of the lamp) and complex inner structures (*i.e.* the seats and windows, etc.). All these results are reported on test data unseen during training time.

As depicted in Fig. 6 and Fig. 1, our HSDF is the first representation that can efficiently reconstruct high-fidelity single-layer open mesh with reasonable face normal directions. These advantages are only possible because 1) HSDFs can model open shapes by hybrid distance and sign; 2) HSDF-Net can learn the prior knowledge of face normal direction in the form of signs, which results in reasonable face directions and also enables using marching cubes to accelerate meshing; 3) Our two-head training manner enables using deep neural networks (known to represent continuous functions) to learn HSDFs. Although we computed HSDFs using surface normal information in training data illustrated in Sec. 4.1, our HSDF-Net is still robust to noise like flipped faces in training data. We can even use the learned HSDF-Net to reconstruct meshes with more reasonable face directions than their ground truth with only point clouds as input, see Fig. 7.

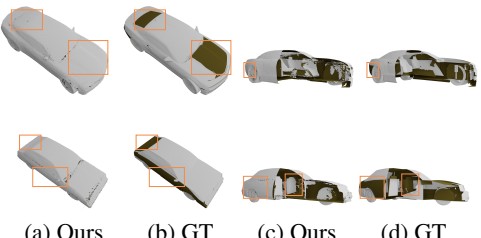

| (a) Ours | (b) GT | (c) Ours | (d) GT |

Figure 7: Reconstructed meshes with face directions corrected. Above are two examples from test data. And they have more reasonable face directions even than their noisy ground truth meshes which have flipped faces. This shows that HSDF-Net learns a pattern of face normal directions from the prior knowledge, which can be robust to small noises.

## 4.5 Further Discussions

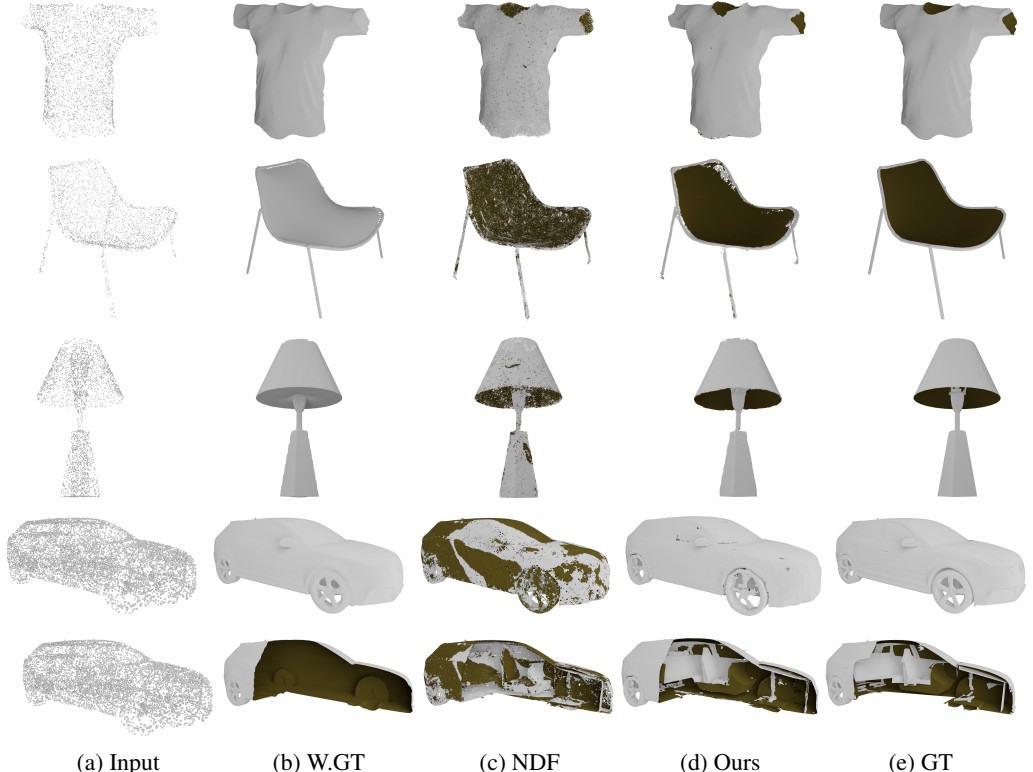

(a) Input      (b) W.GT      (c) NDF      (d) Ours      (e) GT

Figure 6: Shape reconstruction comparison of complex shapes. Dark brown area represents the back-faces. **(a)** input point clouds with 3k points for clothes, chairs and lamps; and 10k points for cars to sample complex inner structures. **(b)** the watertight version of GT pre-processed by DISN [61], which erases the inner structures and closes the open surfaces. This is the upper bound of SDF based approaches. **(c)** reconstructed meshes using BPA [4] on NDFs [15]. The results of NDF suffers because BPA is very sensitive to ball-radius, which have to be tuned per-shape [50]. **(d)** our meshes reconstructed with the proposed HSDF, which preserves the thin shapes as in **(e)** GT.

**Necessity of the two-head training strategy** We validate the necessity of our two-predictor training strategy for sign and distance by jointly training a single decoder to predict a signed distance instead of two decoders. All the results are from test data and reconstructed in the same condition except for the number of decoders. As we can see in column (b) in Fig. 8, although the HSDFs for training are sampled in the same way as two-head training, joint training can never learn HSDFs correctly because a discontinuous function is beyond what a single DNN can express. In contrast, our results in column (c) are a plausible open surface that is similar to the ground truth.

**Necessity of HSDF fusion** We validate the quality improvement by integrating HSDF fusion as described in

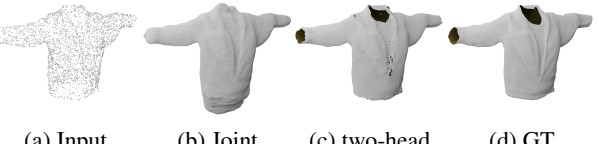

(a) Input      (b) Joint      (c) two-head      (d) GT

Figure 8: Ablation study on two-head training. The reconstructed meshes with a joint learning strategy are shown as column (b) where all the open structures like collars and sleeves are wrongly shaped with watertight surfaces.

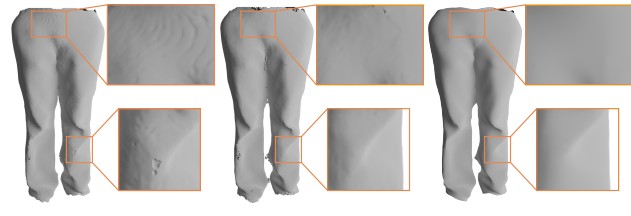

(a) Direct fusing      (b) Grad optimize      (c) GT

Figure 9: Ablation study on sign and distance fusing strategy illustrated in Fig. 3. The gradient optimized HSDF can reconstruct meshes with better surface quality as shown in (b) compared to (a).

Sec. 3.3. As shown in Fig. 9, the gradient optimization consistently removes the Moire pattern and produces smoother results than direct fusing.

**Limitations** The first limitation of our approach is the reconstruction accuracy of mesh boundaries. The marching cube algorithm cuts through every cube to extract the surfaces. The extraction becomes ill-posed when cubes enclose the boundary surface of an open shape, which results in non-manifold structures. We envision that increasing the resolution of boundary cubes will alleviate this problem by decreasing the boundary cubes' size. The second limitation is that we rely on consistent and clean normals of the training data to compute HSDFs. Datasets without clear normals like unprocessed ShapeNet [7] have to be repaired [19] before training.

## 5   Conclusions

We propose a novel hybrid signed and distance field (HSDF) and a dedicated learning model HSDF-Net as an intuitive but useful extension of SDF in previous works. Through sufficient experiments, we have shown that this natural extension leads to a substantial increase in modeling capacities for a much wider range of complex shapes. And we further validate the proposed HSDF can reconstruct arbitrary shapes containing open surfaces efficiently with high fidelity and consistent surface normals. In future work, we expect to explore more applications, such as reconstructions from 2D images to exploit our representational capacities. And we will implement our proposed approach in Jittor [27], which is a fully just-in-time (JIT) compiled deep learning framework.

## 6   Broader Impact

The proposed HSDF representation and the dedicated learning strategy can serve as a fundamental tool for modeling 3D shapes with arbitrary topologies including both open and closed surfaces. Hence, it can have a positive impact to research fields such as computer vision, computer graphics and human-computer interaction, etc. Specifically, due to the enhanced representing and meshing ability of HSDF, our method can reconstruct a wider range of complex shapes with high fidelity and consistent surface normals from raw scanning. This could benefit various real-world applications, including 3D reconstruction tasks from point clouds, images or voxels with thin structures and open surfaces. However, during data collection for our model training, particular care must be taken to ensure that the privacy and security of the owners of private models are not violated.

**Acknowledgments.** This work was supported by CCF-Tencent Open Fund, the Beijing Municipal Natural Science Foundation for Distinguished Young Scholars (No. JQ21013), the National Natural Science Foundation of China (No. 62061136007 and No. 61872440), the Royal Society Newton Advanced Fellowship (No. NAF\R2\192151) and the Youth Innovation Promotion Association CAS.

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
