# HSDF: Hybrid Sign and Distance Field
# for Modeling Surfaces with Arbitrary Topologies
# – *Supplemental Material* –

**Li Wang**[1,2]**, Jie Yang**[1,2]**, Weikai Chen**[3]**,**
**Xiaoxu Meng**[3]**, Bo Yang**[3]**, Jintao Li**[1,2]**, and Lin Gao** (✉)[‡1,2]

[1]Beijing Key Laboratory of Mobile Computing and Pervasive Device, Institute of Computing
Technology, Chinese Academy of Sciences
[2]University of Chinese Academy of Sciences
[3]Tencent Games Digital Content Technology Center
{wangli20s, yangjie01}@ict.ac.cn   chenwk891@gmail.com
{xiaoxumeng,brandonyang}@tencent.com   {jtli, gaolin}@ict.ac.cn

## OVERVIEW

In this paper, we introduce a novel hybrid sign and distance function (HSDF) to model arbitrary shapes including both open and closed surfaces, and design a HSDF-Net with mesh extraction algorithm to obtain single-layer meshes efficiently. This supplementary material consists of some more detailed analysis and experimental visualization results, which accompanies the main paper.

All the sections are organized as follows:

- Section 1.1 provides a detailed network architecture of our proposed HSDF-Net.

- Section 1.2 illustrates that the process of calculating the sign and distance of the sampled points according to the arbitrary surface, in terms of our novel representation HSDF.

- Section 1.3 and Section 1.4 provide more details on network implementation and mesh extraction.

- Section 2 provides additional qualitative comparisons for meshes reconstructed from sparse point clouds.

## 1 Implementation Details

### 1.1 Network Architecture

Our network consists of a multi-scale volume encoder, a sign predictor, and a distance predictor. The multi-scale volume encoder is adopted from IF-Net [4], which extracts features from voxelized sparse point clouds via 3D convolution and max-pooling. The detailed network architecture is listed in Table 1. Since the feature maps of 3D convolution have a regular grid form, we can easily interpolate these feature grids at the positions of query points to obtain the specific features of query points. Then we can obtain the interpolated features of query points in a multi-scale fashion, which intrinsically are encoded to aggregate the shape feature at different levels. More specifically, for every single point, we have six shifted points with a displacement of 0.0722 along the x, y, and z-axis respectively, see IF-Net [4]. And we merge the seven points (including itself) into a group to trilinearly interpolate the features from different levels of feature grids.

---

*Corresponding author is Lin Gao (gaolin@ict.ac.cn).

36th Conference on Neural Information Processing Systems (NeurIPS 2022).

| Layer | Kernel size | Stride | Padding | Activation function | $(C_{out}, D, H, W)$ | Grid Sample |
|---|---|---|---|---|---|---|
| Input voxels | - | - | - | - | (B,1,256,256,256) | (B,1,7,P) |
| Conv3d | (3,3,3) | (1,1,1) | (1,1,1) | ReLU,BN | (B,16,256,256,256) | (B,16,7,P) |
| MaxPool3d | (2,2,2) | (2,2,2) | (0,0,0) | - | (B,16,128,128,128) | - |
| Conv3d | (3,3,3) | (1,1,1) | (1,1,1) | ReLU | (B,32,128,128,128) | - |
| Conv3d | (3,3,3) | (1,1,1) | (1,1,1) | ReLU,BN | (B,32,128,128,128) | (B,32,7,P) |
| MaxPool3d | (2,2,2) | (2,2,2) | (0,0,0) | - | (B,32,64,64,64) | - |
| Conv3d | (3,3,3) | (1,1,1) | (1,1,1) | ReLU | (B,64,64,64,64) | - |
| Conv3d | (3,3,3) | (1,1,1) | (1,1,1) | ReLU,BN | (B,64,64,64,64) | (B,64,7,P) |
| MaxPool3d | (2,2,2) | (2,2,2) | (0,0,0) | - | (B,64,32,32,32) | - |
| Conv3d | (3,3,3) | (1,1,1) | (1,1,1) | ReLU | (B,128,32,32,32) | - |
| Conv3d | (3,3,3) | (1,1,1) | (1,1,1) | ReLU,BN | (B,128,32,32,32) | (B,128,7,P) |
| MaxPool3d | (2,2,2) | (2,2,2) | (0,0,0) | - | (B,128,16,16,16) | - |
| Conv3d | (3,3,3) | (1,1,1) | (1,1,1) | ReLU | (B,128,16,16,16) | - |
| Conv3d | (3,3,3) | (1,1,1) | (1,1,1) | ReLU,BN | (B,128,16,16,16) | (B,128,7,P) |
| MaxPool3d | (2,2,2) | (2,2,2) | (0,0,0) | - | (B,128,8,8,8) | - |
| Conv3d | (3,3,3) | (1,1,1) | (1,1,1) | ReLU | (B,128,8,8,8) | - |
| Conv3d | (3,3,3) | (1,1,1) | (1,1,1) | ReLU,BN | (B,128,8,8,8) | (B,128,7,P) |

Table 1: The network architecture of volume encoder where B is the mini-batch number and P is the sub-sample number of query points. Grid sampling is a trilinear interpolation of feature grids at query points' positions. The last two dimensions (7, P) of grid sampling output means that every query point and its six extra neighbor points form a group to get combined interpolated features.

| Layer | $H_{in}$ | Activation function | $H_{out}$ |
|---|---|---|---|
| shape code + (x,y,z) | (3479+3) | - | (3482) |
| fully-connected | (3482) | ReLU | (512) |
| fully-connected | (512) | ReLU | (256) |
| fully-connected | (256) | ReLU | (256) |
| fully-connected | (256) | ReLU | (1) |

Table 2: The network architecture of distance predictor.

The size of output feature is shown in Table 1 (Column 'Grid Sample'). After the multi-scale feature interpolation, we concatenate all the features interpolated from different feature grids levels, so the concatenated feature's channel number is $(1 + 16+32+64+128+128+128) \times 7 = 3479$.

| Layer | $H_{in}$ | Activation function | $H_{out}$ |
|---|---|---|---|
| shape code + (x,y,z) | (3479+3) | - | (3482) |
| fully-connected | (3482) | ReLU | (512) |
| fully-connected | (512) | ReLU | (256) |
| fully-connected | (256) | ReLU | (256) |
| fully-connected | (256) | - | (1) |

Table 3: The network architecture of sign predictor.

The detailed architecture of our sign predictor and distance predictor are presented in Table 3 and Table 2. The sign predictor and distance predictor take the above output features of the encoder as input, the channel number of the input feature is $3479 + 3 = 3482$. Finally, the distance predictor follows a ReLU activation to ensure the predicted distance is a non-negative value.

## 1.2 Data Preparation

A 2D open shape $\mathcal{S}$ is depicted in Fig. 2, point A and point B are the sampled point in the space. We denote the position of A as $p_A$ and its closest point $A'$ as $p_{A'}$ to the surface $\mathcal{S}$. The surface normal at $A'$ is $n_{A'}$, then we can decide the sign of point A by the sign of dot product of $dot(p_A - p_{A'}, n_{A'})$, where $dot(\cdot, \cdot)$ means a dot product of two vectors. If $dot(p_A - p_{A'}, n_{A'}) > 0$, then we assign the sign of A to be positive and vise versa for point B. Even though point B is close to the disconnected region of the open shape, its sign can still be computed in this way. In Fig. 1, we show some visualized results of computed signs of open shapes where the blue region indicates negative signs and the red region indicates positive signs. We prepare our training data by sampling points near the surfaces of any shapes, following NDF [5]. More specifically, we sample 300,000 points per model during data

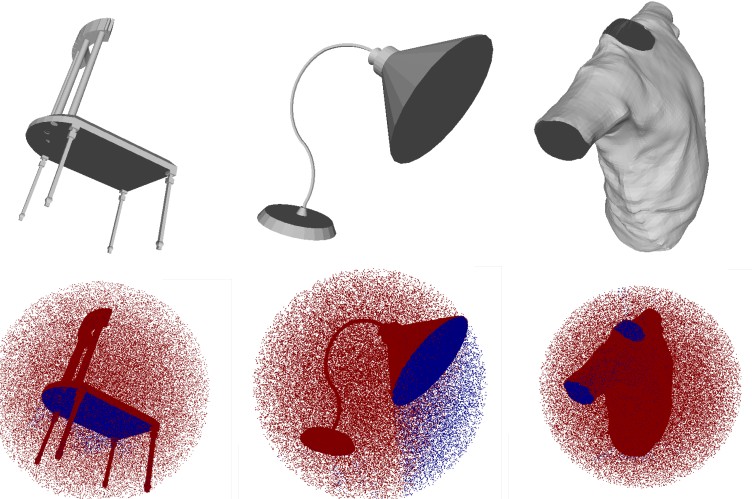

Figure 1: Computing visualization of HSDF . Shapes in the first row are some examples of open shapes, and figures in the second row are the sign visualization of their computed HSDFs.

preparation. We use a subset of the sample points that contains 90,000 points during training time, and $1\%$ of samples are within distance 0.08, $49\%$ of samples are within distance 0.02, and $50\%$ are within distance 0.003. Sampling near the surface allows us to approximate more detailed geometry and boundary of arbitrary surfaces and contributes to less ambiguous signs.

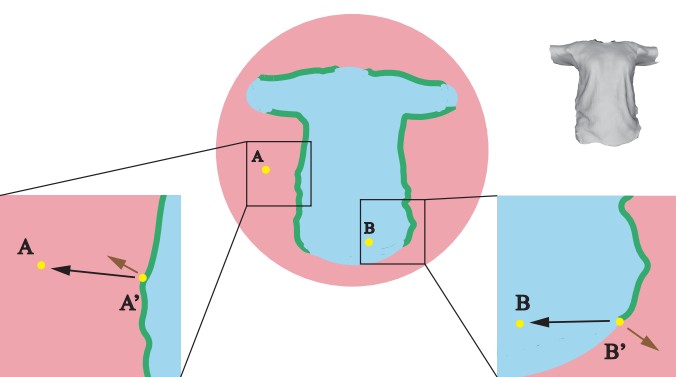

Figure 2: A 2D illustration of sign computing for open surfaces. Green contour depicts an open shape in 2D and blue, red region indicates positive, negative signs respectively. For point A, we denote its closest surface point as $A'$ and define their coordinates as $p_A$ and $p_{A'}$. The black vector represents $p_A - p_{A'}$ while the brown vector represents surface normal $v$ at $A'$. Thus if dot product $v \cdot (p_A - p_{A'})$ is positive as the case for point A, we assign its sign to be positive and vice versa for point B.

## 1.3  Network Training and Inference

We implement the proposed HSDF-Net in PyTorch [8]. And we use the Adam optimizer with a learning rate of $1e-4$ to optimize the trainable parameters of our network. The parameters of our volume encoder are initialized to the pre-trained weights released by the author of NDF [5] to make our training converge fast. The sign predictor and distance predictor is initialized by kaiming uniform initialization [7]. During training, we set the batch size as 2 and use 90,000 points (sub-set) per shape. We trained the models until both the validation minimum of the distance predictor and the best validation accuracy of the sign predictor are reached. Our network can converge in about 70 epochs in 2 days with a single NVIDIA GeForce RTX 3090 GPU and an Intel i5-9600K CPU.

The dataset is split into training, validation, and test set with the ratio of 0.7, 0.1, and 0.2. For all our experiments, we sample 3000 sparse point clouds as input for fair comparisons.

## 1.4  Mesh Extraction

For mesh extraction, the resolution is 256 for our 3D HSDF grid, and we extract the explicit surfaces by our masked Marching Cubes with a zero threshold. In terms of NDF [5], we use the released code provided by the authors to generate a dense point cloud with 1M points and reconstruct it as mesh using the Ball-Pivoting algorithm (BPA) [1] in MeshLab [6] with a ball radius of 0.01. This ball radius is also adopted by [9]. As we demonstrated in our main paper, we have found the cost of computation and visual quality of results by BPA is highly sensitive to the ball-radius threshold. In many cases, the threshold must be adjusted manually for each reconstructed shape. In CSP-Net [9], it also reveals the drawbacks of their experiments.

In contrast, our HSDF simply reconstructs arbitrary surface (*i.e.* open and close surfaces) by a masked Marching Cubes algorithm with better visual quality and only take a shorter time than the BPA process for the shape reconstruction from sparse point clouds.

## 2  More Additional Visualization Results

In this section, we provide more additional visualization results, which are shown in Fig. 3, Fig. 4, Fig. 5, Fig. 6, Fig. 7, Fig. 8, and Fig. 9. For all the displayed results, we compare with two kinds of mesh extractions from NDF [5]. We provide representative results containing thin-shell open shapes and complex inner structures, which further demonstrate that our proposed HSDF is the first shape representation that is able to reconstruct high fidelity open and close shapes with the more accurate surface normal efficiently. All the results are reconstructed from test data, which is unseen during training time. From the visualization results, it is obvious that the performance of shape reconstruction from sparse point clouds outperforms the alternative SOTA approaches, such as NDF [5].

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

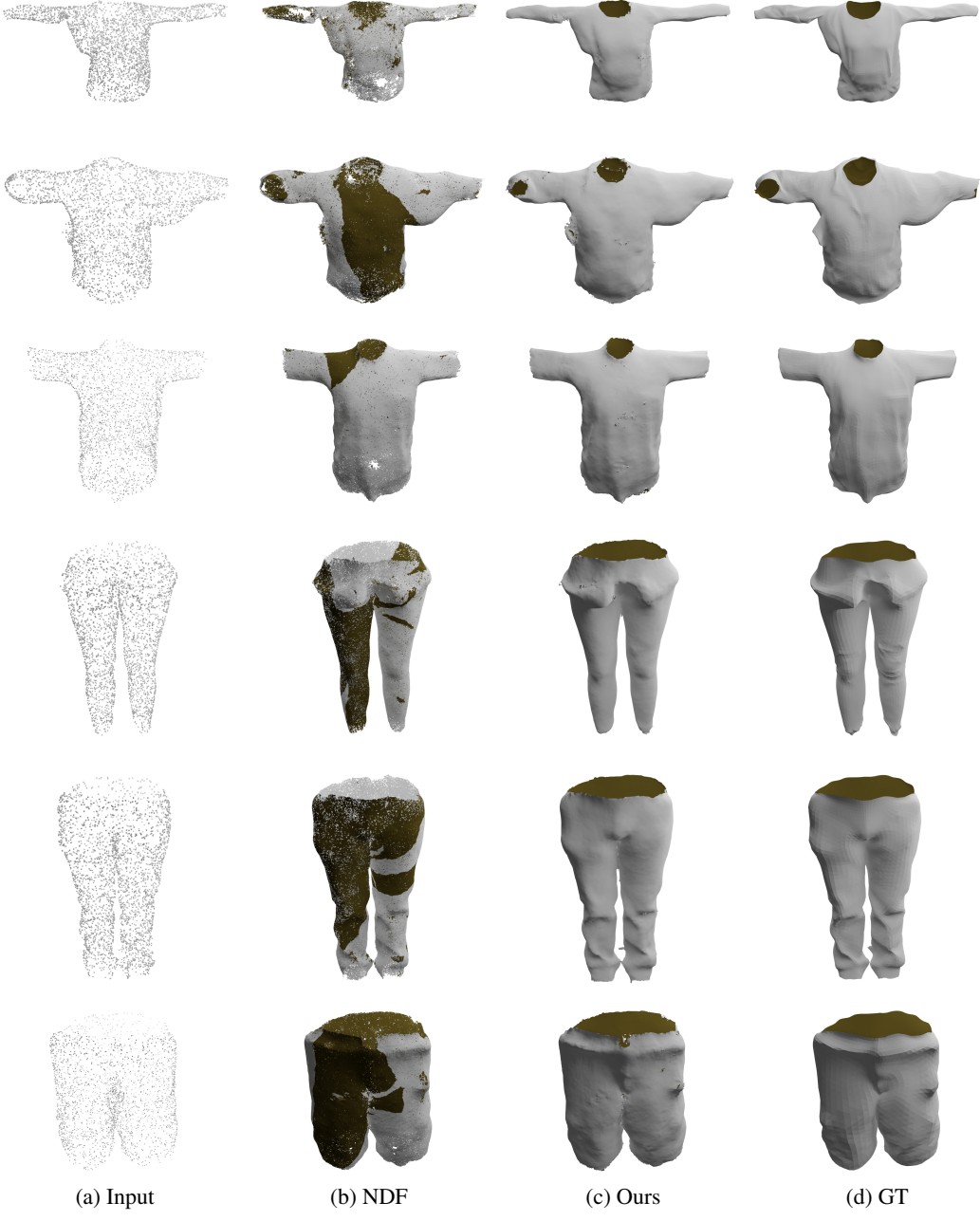

(a) Input        (b) NDF        (c) Ours        (d) GT

Figure 3: Additional comparisons on MGN dataset [2]

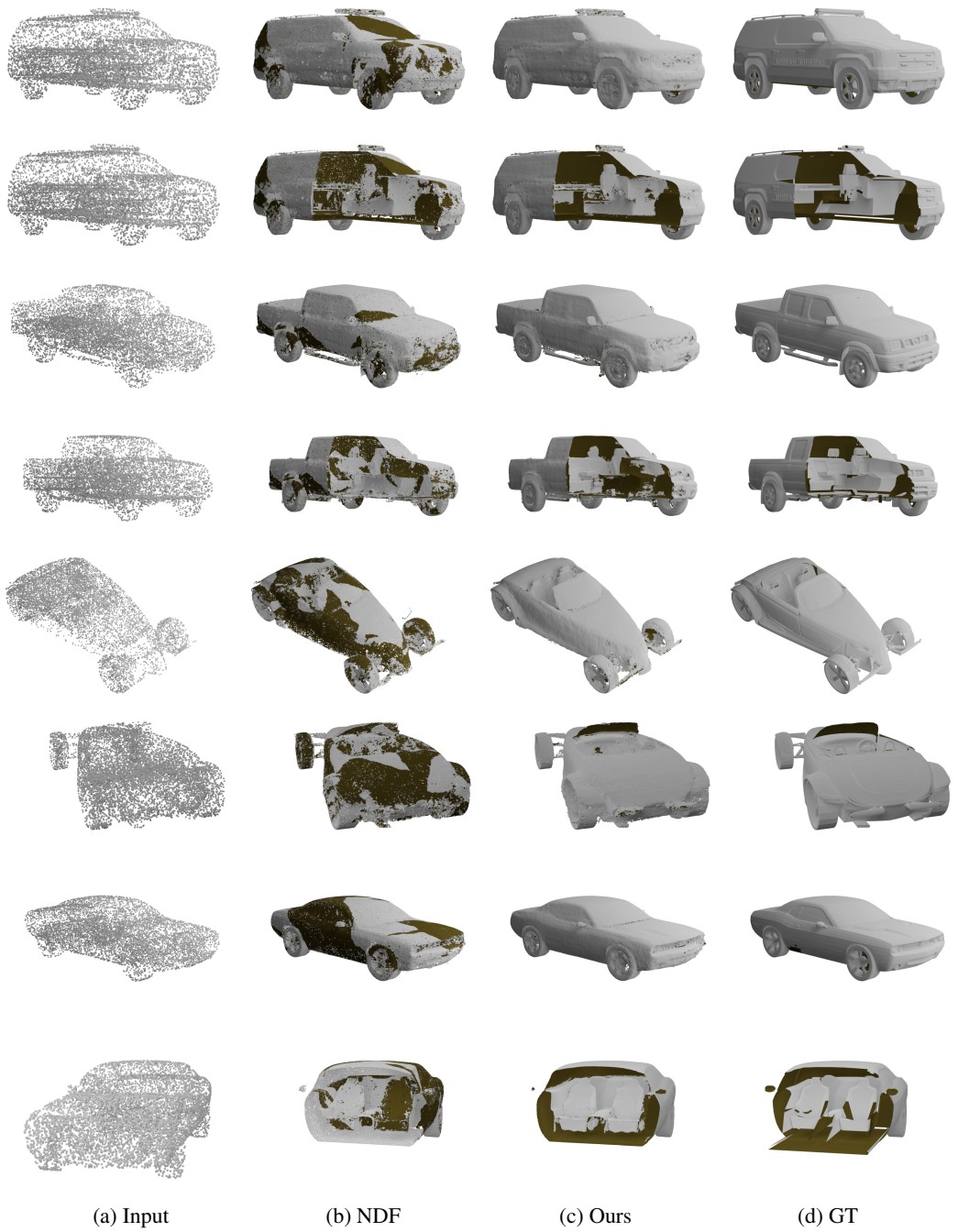

(a) Input      (b) NDF      (c) Ours      (d) GT

Figure 4: Additional results on ShapeNet [3] cars.

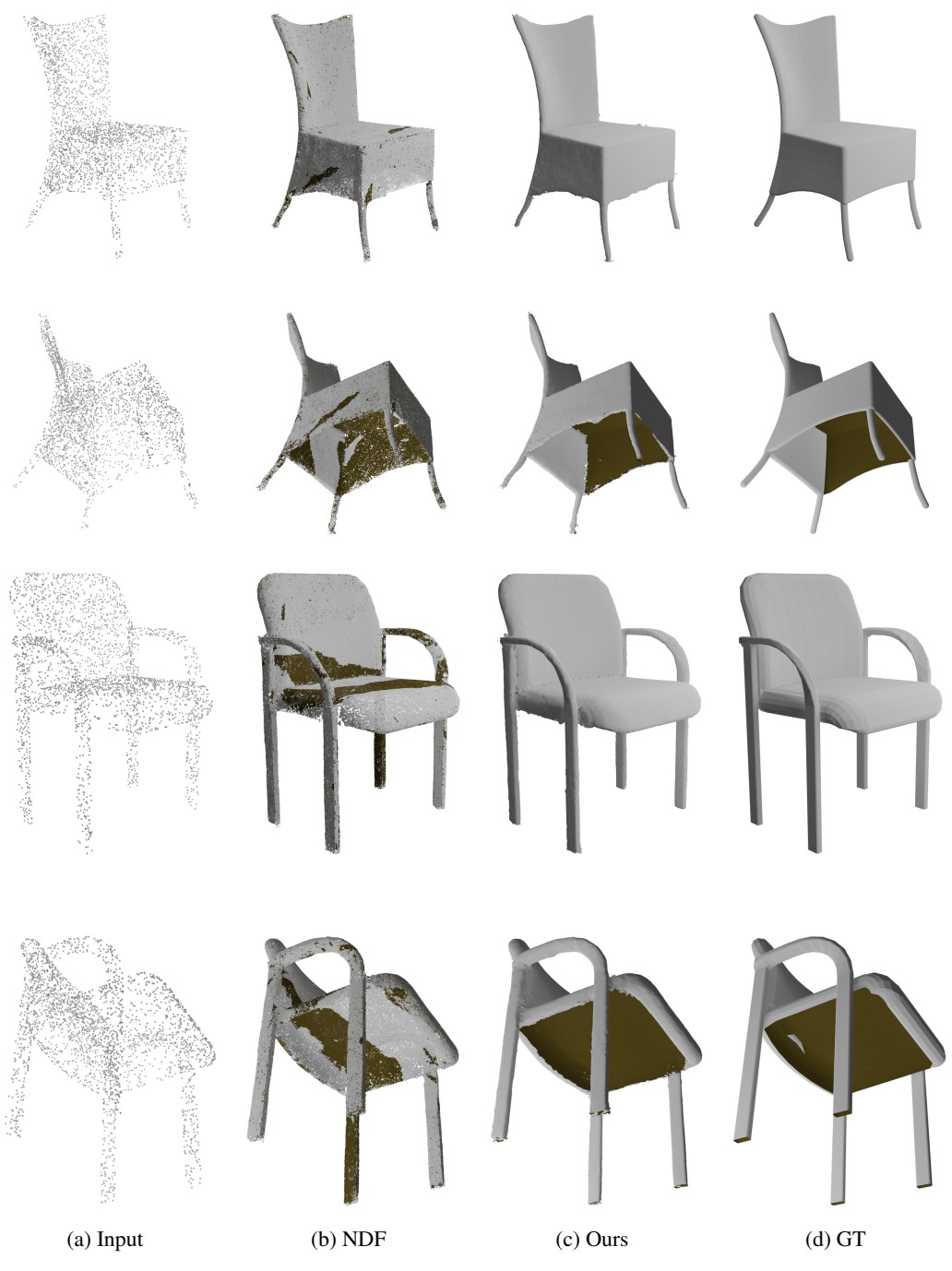

(a) Input      (b) NDF      (c) Ours      (d) GT

Figure 5: Additional results on ShapeNet [3] chairs.

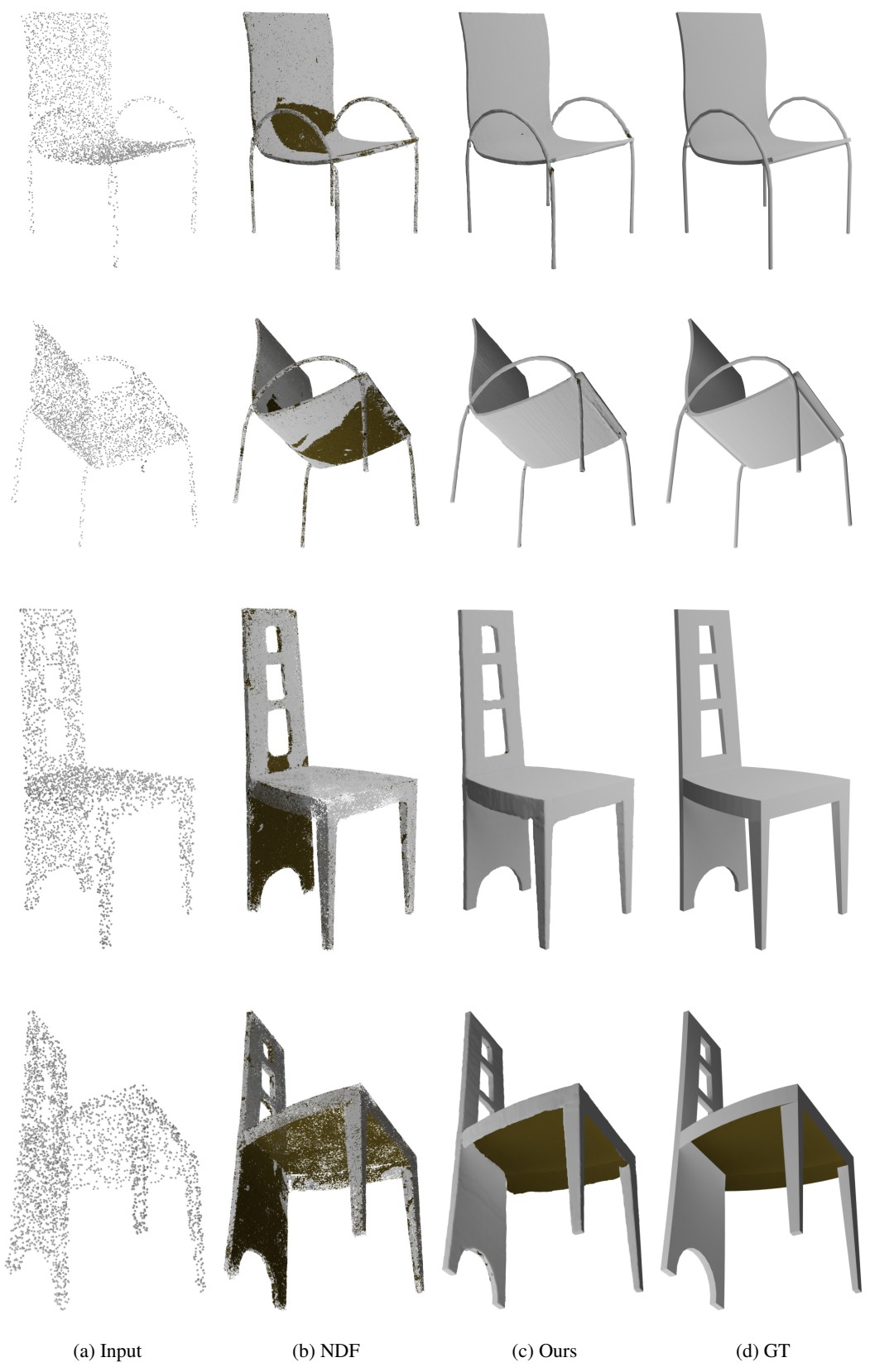

(a) Input      (b) NDF      (c) Ours      (d) GT

Figure 6: More additional results on ShapeNet [3] chairs.

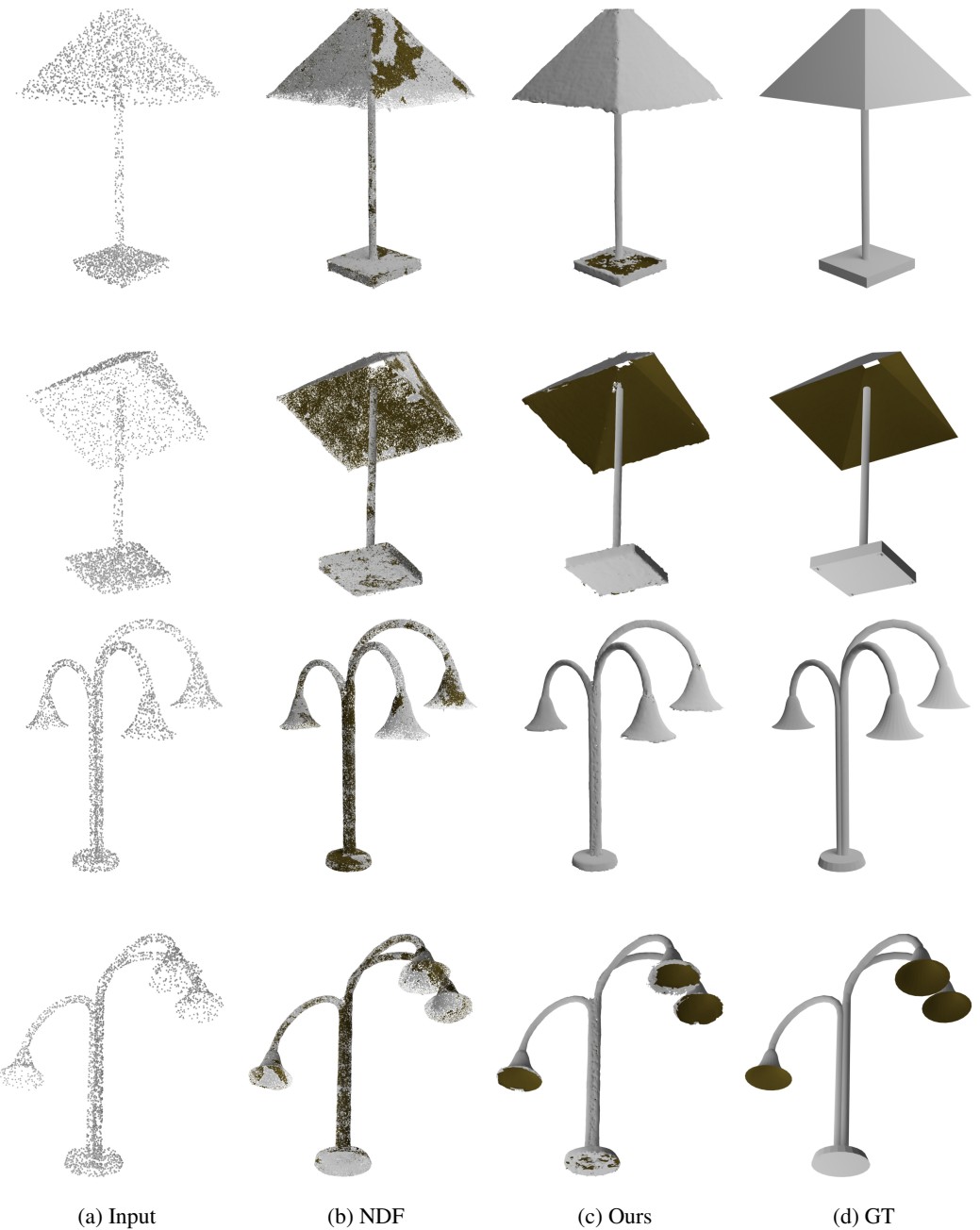

(a) Input         (b) NDF         (c) Ours         (d) GT

Figure 7: Additional results on ShapeNet [3] lamps.

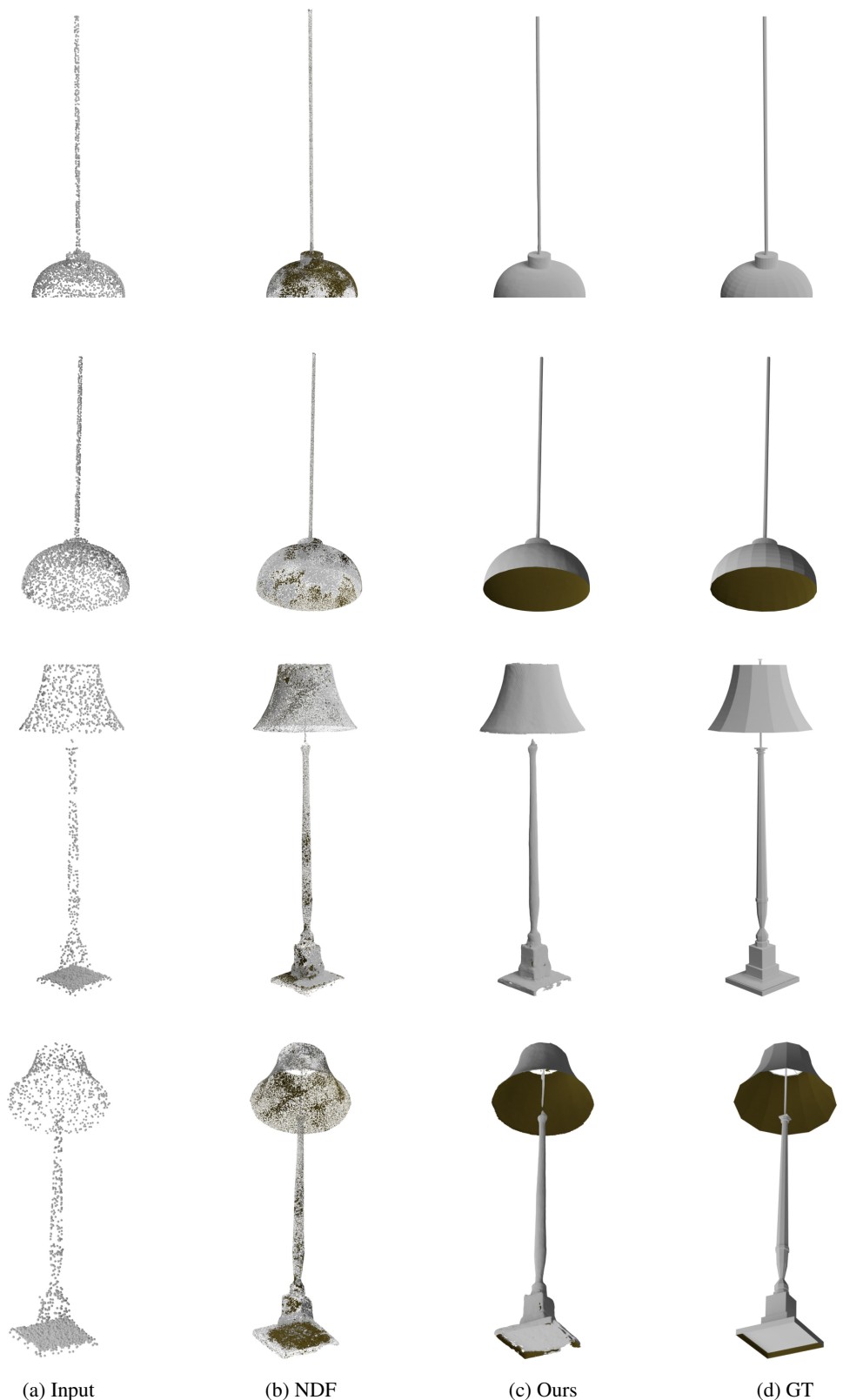

(a) Input       (b) NDF       (c) Ours       (d) GT

Figure 8: More additional results on ShapeNet [3] lamps.

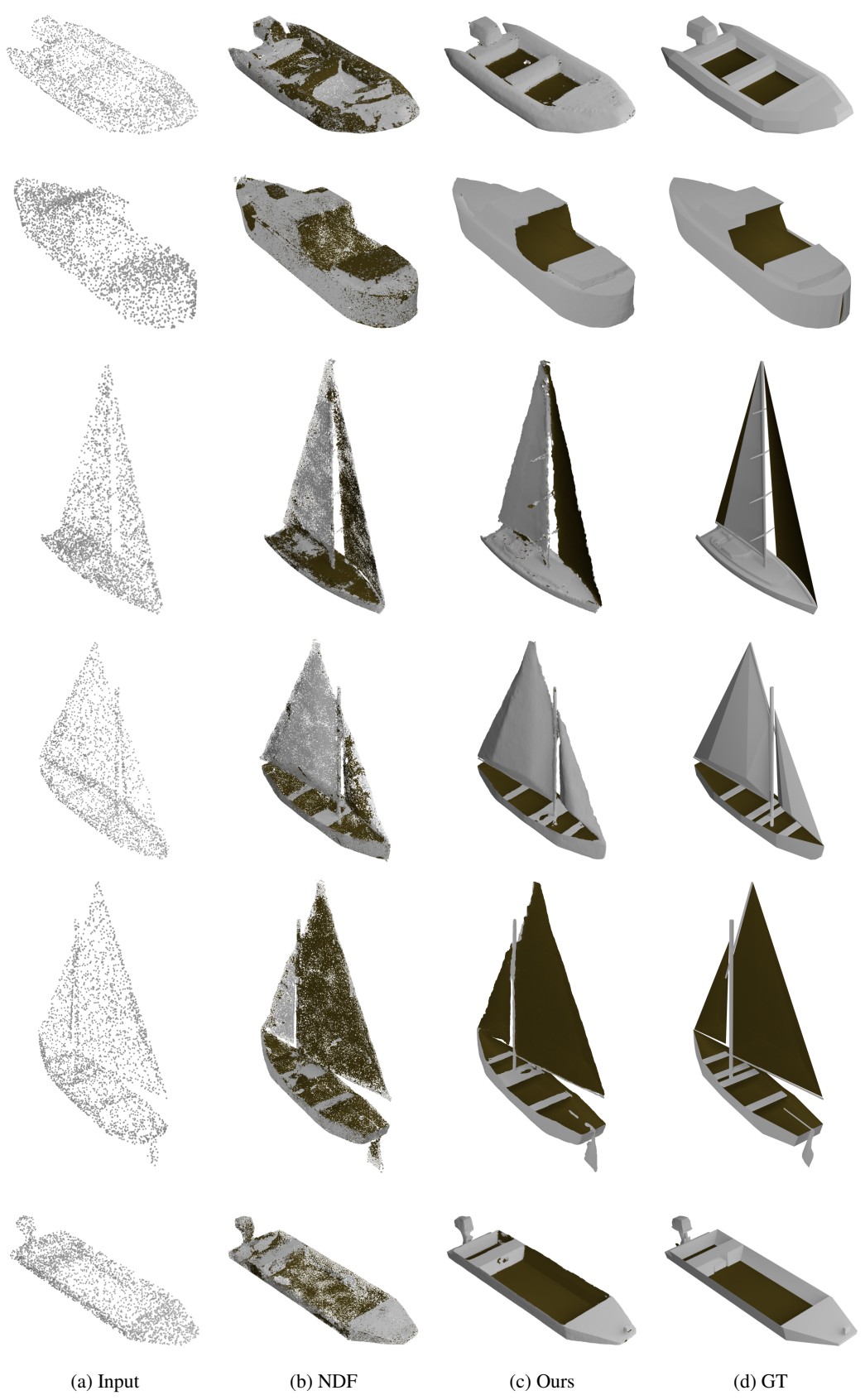

(a) Input       (b) NDF       (c) Ours       (d) GT

Figure 9: Additional results on ShapeNet [3] ships.