# OpenReview forum: "HSDF: Hybrid Sign and Distance Field for Modeling Surfaces with Arbitrary Topologies"
_NeurIPS.cc/2022/Conference — NeurIPS 2022 Accept_

### Official Review · Reviewer_gpXi · 2022-07-07

**Rating:** 5
**Confidence:** 4
**Soundness:** 3 good
**Presentation:** 3 good
**Contribution:** 3 good

**Summary:**

This paper proposes a hybrid distance field approach to represent 3D object surfaces, called HSDF. The main idea behind it is to utilize the advantage of the unsigned distance function and use another network head to predict the sign of the distance field for each point. The network structure is inherited from IF-net with additional heads. To extract the mesh from HSDF, the authors propose a masked Marching Cube algorithm to extract the iso-surface from a union of the local bounding boxes to handle the arbitrary topology of the 3D shape. The experiments conducted in ShapeNet demonstrate the HSDF achieves good performance compared with other methods.

**Questions:**

1. Several recent works with similar ideas exist, such as [A]. It would be better if the authors could discuss and compare the differences.

2. Discussion about robustness. The input of all comparison methods is a point cloud. Suppose the point cloud contains some unexpected noise, what about the performance drop or influenced by the perturbation of inputs?

3. The notation in this paper might need further improvement.

    a. $Sign$ in line 192 is a signed distance function, I think it should be $\text{HSDF}(\mathbf{p}, \mathbf{X}) = \text{sign}(Sign_\mathbf{x}(\mathbf{p}))*Dis_\mathbf{x}(\mathbf{p})$ in line 213.

    b. line 184 and line 186, the mapping should be a binary output {-1, 1} rather than an interval $[-1, 1]$?



[A] 3PSDF: Three-Pole Signed Distance Function for Learning Surfaces with Arbitrary Topologies (CVPR'22)



**Ethics Review Area:**

["I don’t know"]

**Limitations:**

The authors have discussed the limitation adequately.

**Strengths And Weaknesses:**

1. The design of HSDF is simple but effective. The experimental result and ablation study also demonstrate its effectiveness.
2. The motivation of the proposed approach is reasonable and the idea is easy to follow.
3. The gradient optimization idea in Sec 3.3 is thought-provoking.

---

> ### Author Response · Authors · 2022-08-02
> **Response to Reviewer gpXi**
>
> We thank the reviewer for the insightful comments and suggestions! We provide our detailed response below.
>
> > Q1: Several recent works with similar ideas exist, such as [A]. It would be better if the authors could discuss and compare the differences.
>
> A1: Our work is concurrent to 3PSDF[A]. 3PSDF proposes an implicit representation that divides the space into three classes, namely positive, negative and null. Compared to watertight SDF, their null class enables 3PSDF to mask out certain regions in the space and represent open shapes. 3PSDF formulates the reconstruction task as a classification problem. It eases the learning difficulty but also limits the applications on downstream tasks that require SDF continuity, such as neural rendering. In contrast, HSDF can be adapted for neural rendering thanks to its continuous nature. Since they released their paper and code after our submission deadline, we didn't compare them in our main paper. We will add discussions with 3PSDF in our revision.
>
> [A] 3PSDF: Three-Pole Signed Distance Function for Learning Surfaces with Arbitrary Topologies (CVPR'22)
>
>
> >Q2: Discussion about robustness. The input of all comparison methods is a point cloud. Suppose the point cloud contains some unexpected noise, what about the performance drop or influenced by the perturbation of inputs?
>
> A2: We have evaluated it on the MGN test set. For every point $p$ in the input point cloud, the noisy $p'$ is calculated by the formulation $p'=p + \sigma*(dx, dy, dz)^T$, where dx, dy and dz are sampled from standard normal distribution $N(0,1)$ and $\sigma$ is a coefficient that describes the level of noise. And we use the pre-trained model checkpoint which is the same as our main paper. The evaluations on chamfer distance (x$10^{-4}$) are shown below:
> ||$\sigma=0$|$\sigma=0.005$|$\sigma=0.01$|$\sigma=0.02$|
> |:-:|:-:|:-:|:-:|:-:|
> |NDF|0.158|0.203|0.247|0.376|
> |Ours|**0.151**|**0.199**|**0.242**|**0.362**|
>
> We will add more robustness comparisons to our revision.
>
> > Q3: The notation in this paper might need further improvement.
>
> A3: Thank you for the advice! We will improve the notations in our revision.

---

### Official Review · Reviewer_nGbx · 2022-07-11

**Rating:** 5
**Confidence:** 4
**Soundness:** 3 good
**Presentation:** 3 good
**Contribution:** 2 fair

**Summary:**

As opposed to SDFs, this submission proposes to regress sign and distance separately, and extend sign to oriented open (ie. non watertight) meshes. Authors demonstrate it can represent both open and close surfaces, and propose a way to mesh it and merge sign and distance fields that reduces inaccuracies of both components.


**Questions:**

- In 3.1, how is the surface normal v computed? does it assume the surface is meshed, and uses triangle winding numbers? This is a crucial aspect of the submission that should be made clearer.
- Why is the propose "sign and distance fusion" procedure required? Which field is noisy? Both?
- Is Tab. 1 measuring meshing time only? or both network forward pass and meshing time? Is NDF using ball pivoting here?
- Loss term L_s : why clamp a binary sign value? I guess there is a typo here, what is the real loss term?
- While the introduction focuses on the generative/representation aspect of 3D shapes, the method section starts with an input point cloud. Could the shape representation be demonstrated in other settings, like auto-decoder?
- What are the "adversarial impacts" of SDF and UDF mentioned in the introduction?

**Limitations:**

Authors have identified mesh boundaries and consistent normal orientations as main limitations, but do not quantify it. For example:
- would a metric computed only on mesh borders really show that boundaries are not well represented?
- across the main 3D datasets, which ones have consistent normals? What are pathological cases (Moebius strip)?

**Strengths And Weaknesses:**

Strengths:
- In the experiment section, metrics include both a surface proximity (CD) and smoothness (NC) components, and are in favor of the proposed approach agains NDF.
- Results look visually appealing.
- Ablations demonstrate the effectiveness of the sign+distance fusion.


Weaknesses:
- The main limitation is that learning a prior on triangle facets orientation requires datasets to be properly oriented, which might not always be possible. This huge limitation is only slightly mentioned in Fig. 7 but should be discussed more thoroughly and sooner.
- The current formulation reads as if factoring SDFs into distance times sign was making the field easier to be learned by MLPs because it is more continuous. But in fact the sign network is highly non continuous.
- The intro states that "..." , but this is not true, see NDF (Chibane 2020) for example. The proposed "Masked Marching Cubes" could be applied to normal SDFs, and should also be demonstrated in this case.
- Wording: I do not find the word "hybrid" well chosen to describe the proposed representation. "Factorised" or "Disentangled" would be more appropriate.
- The first 3 lines of the introduction only mention SDFs and forget about Occupancy Fields.
- The "sign and distance fusion" is not motivated by any intuition.
- Tab.2 evaluates oriented normal consistency for methods that are not supervised for it. This is a biased assessment.
- Some references about representing/meshing open surfaces with UDFs are missing, such as "Neural Dual Contouring" or "MeshUDF". "DUDE" is mentioned, but not used as a baseline in the experiment section.
- English quality could be improved.

Minor: figures always appear 1 page before their reference in the text.

---

> ### Author Response · Authors · 2022-08-02
> **Response (2/2) to Reviewer nGbx**
>
> > Q6: Why is the propose "sign and distance fusion" procedure required? Which field is noisy? Both? The "sign and distance fusion" is not motivated by any intuition.
>
> A6: Neither of the fields is noisy. We introduce fusion optimization as an **"registration of the zero level sets of the distance and sign fields"**. We have shown this ablation study in Figure 9. The naive fusion of sign and distance is simply multiplying them. However, due to the separate learning strategy, the zero level set of the sign field won't be perfectly aligned with that of the distance field. This would slightly reduce the surface quality. Hence, we propose this fusion optimization to register these two zero level sets using the first-order information.
>
> > Q7: Tab.2 evaluates oriented normal consistency for methods that are not supervised for it. This is a biased assessment.
>
> A7: We use the oriented normal consistency (NC) to demonstrate that our method is able to reconstruct more consistent and accurate face normals (including their directions) for open shapes, where the other baseline method can't. To alleviate the concern of biased assessment, we provide additional evaluations on the original version of NC in the table below. All the results are reported on the test set with 3000 points as input. Due to the poor mesh surface quality reconstructed from dense point clouds using BPA, our method still outperforms NDF on the original NC metric.
>
> ||MGN|Car|Chair|Ship|Lamp|Mean
> |:-:|:-:|:-:|:-:|:-:|:-:|:-:|
> |NDF|0.938|0.817|0.865|0.850|0.869|0.868
> |Ours|**0.957**|**0.842**|**0.896**|**0.863**|**0.904**|**0.892**
>
> We will add this result to the revision.
>
> > Q8: Some references about representing/meshing open surfaces with UDFs are missing, such as "Neural Dual Contouring" or "MeshUDF". "DUDE" is mentioned, but not used as a baseline in the experiment section.
>
> A8: We will add these missing references in the revision. "DUDE" hasn't released its source code. Hence, we use NDF (source code available) as our baseline to ensure fair comparison. If requested, we will implement "DUDE" in the revision to provide additional comparisons.
>
> > Q9: English quality could be improved.
>
> A9: Thank you for the advice! We will improve our writing in the revision.
>
> ## Minor Issues
>
> We thank the reviewer for the constructive detailed feedback concerning the mathmatical symbols, unclear statements, and detailed implementation of our work. The well-documented code and pre-processed data will also be released to assist the readers to reproduce and understand our algorithm deeply.
>
> > Figures always appear 1 page before their reference in the text.
>
> We will fix it in the revision.
>
> > In 3.1, how is the surface normal v computed? does it assume the surface is meshed, and uses triangle winding numbers? This is a crucial aspect of the submission that should be made clearer.
>
> The detailed computing process is introudced in Section 4.1 of the HSDF computation paragraph. We will add more details on normal computation to Section 3.1 in our revision.
>
> > Is Tab. 1 measuring meshing time only? or both network forward pass and meshing time? Is NDF using ball pivoting here?
>
> We report the whole inference time including both forward pass and meshing time here. For the meshing of NDF, we use the MeshLab scripts released by the NDF's author to apply the BPA.
>
> > Loss term L_s : why clamp a binary sign value? I guess there is a typo here, what is the real loss term?
>
> Here the function **Sign(*)** returns a continuous value. We provide its definition in line 183. But it is indeed a typo -- there should be a two-directional clamp instead of a one-directional clamp. The correct loss term is rewritten below. We will fix this typo in our revision.
>
> $L_s = \sum\limits_{x\in B}\sum\limits_{p\in P}|max(min(Sign_x(p),\delta),-\delta) - max(min(SDF(p, S_x),\delta),-\delta)|$
>
> > While the introduction focuses on the generative/representation aspect of 3D shapes, the method section starts with an input point cloud. Could the shape representation be demonstrated in other settings, like auto-decoder?
>
> Yes, HSDF is a general implicit representation that can scale to other application settings, including auto-decoder. Specifically, both the sign and distance fields can be learned in a auto-decoding fashion. We use sparse point cloud as input following the experiment settings of NDF mainly to ensure fair comparisons. We will explore more interesting applications in the future.
>
> > What are the "adversarial impacts" of SDF and UDF mentioned in the introduction?
>
> We mean that their respective shortcomings demonstrated in the first 2 paragraphs of the introduction, which is: SDF can only represent watertight surfaces while UDF suffers from the meshing quality problem when converting the point cloud to mesh.

---

> > ### Comment · Reviewer_nGbx · 2022-08-08
> > **Thank you for your answers.**
> >
> > Two minor points:
> > - I would avoid using the word "adversarial" in the context of deep neural networks. "Respective shortcomings" is much more precise.
> > - By "..." I meant "Unlike traditional SDF, HSDF is able to locate the surface of interest". This is not a novelty of HSDF, NDF introduced it and it can be applied to any SDF.

---

> > > ### Author Response · Authors · 2022-08-09
> > > **Thank you for your suggestive comments.**
> > >
> > > We want to thank you for recognizing our work and for your insightful advice! We really appreciate that! And we will improve these wordings and expressions as you suggested. Thank you!:-)

---

> ### Author Response · Authors · 2022-08-02
> **Response (1/2) to Reviewer nGbx**
>
> We thank the reviewer for the insightful comments and suggestions! We provide our detailed response below.
>
> ## Main Questions
>
> > Q1: The main limitation is that learning a prior on triangle facets orientation requires datasets to be properly oriented, which might not always be possible. This huge limitation is only slightly mentioned in Fig. 7 but should be discussed more thoroughly and sooner.
>
> A1: As we mentioned in Section 4.1, all the training data can be robustly oriented using the released code of [A]. Another concurrent work, 3PSDF [B], also uses this work [A] for consistent normal orientation and achieves impressive results. Once the surface normal of training data is pre-processed, our method does not require any further processing for the test data. We will release our pre-processed data to encourage future research. In addition, more discussion on data processing will be discussed more thoroughly and earlier in the revision.
>
> [A] Repairing man-made meshes via visual-driven global optimization with minimum intrusion (SIGGRAPH Asia 2019)
>
> [B] 3PSDF: Three-Pole Signed Distance Function for Learning Surfaces with Arbitrary Topologies (CVPR'22)
>
> > Q2: The current formulation reads as if factoring SDFs into distance times sign was making the field easier to be learned by MLPs because it is more continuous. But in fact, the sign network is highly non-continuous.
>
> A2: We agree that the sign field is not continuous for open surfaces. However, **the discontinuity only happens in the regions that are far from the surface**. Our approach, on the other hand, only needs to learn an accurate sign field in the **vicinity of the surface**, as our distance field will guide our reconstruction to it (Section 3.4). In fact, we leverage an importance sampling strategy to focus our training samples around the target surface (Section 1.2 of the supplemental materials) -- 99% of samples are distributed in a narrow band around the surface where the sign field is highly continuous. Hence, our sign field is easy to learn.
>
>
> In addition, in Figure 8, we provide an ablation study comparing the performance of an alternative method that does not factorize the SDF into sign and distance fields. The discontinuity of the sign field far from the surface increases the difficulty of joint learning and leads to erroneous reconstruction. This further verifies the effectiveness of our factorized learning. We will make this clearer in our revision.
>
>
> > Q3: The intro states that "..." , but this is not true, see NDF (Chibane 2020) for example. The proposed "Masked Marching Cubes" could be applied to normal SDFs, and should also be demonstrated in this case.
>
> A3: We are not sure what the "..." refers to. Here, we can only provide our response based on our limited understanding. We agree that the proposed "Masked Marching Cubes" can be applied to the normal SDFs to cast shapes with arbitrary topologies. However, the key question to ask is how to obtain such a mask only from a normal SDF. In our method, we leverage the gradient field of UDF to push a query point to its closest neighbor on the zero-level surface. The grid cell that the query point finally lands on is masked as valid for performing marching cubes (mesh extraction). However, for a normal SDF, as it can only represent a closed surface, it remains an open question how to extract a mask solely based on SDF such that it can help reconstruct an open surface that matches one's goal. Specifically, even we compute an unsigned field from an SDF by calculating its absolute value, the mask we obtain from its gradient field remains a closed surface. We are happy to provide more demonstrations if clearer instruction on how to obtain a mask from a normal SDF is provided.
>
>
> > Q4: Wording: I do not find the word "hybrid" well chosen to describe the proposed representation. "Factorised" or "Disentangled" would be more appropriate.
>
> A4: Thank you for your advice! We will consider changing the wording in the revision.
>
> > Q5: The first 3 lines of the introduction only mention SDFs and forget about Occupancy Fields.
>
> A5: We introduced Occupancy Fields in our related works, and we will also add this to our introduction in our revision.

---

### Official Review · Reviewer_oe6G · 2022-07-11

**Rating:** 5
**Confidence:** 4
**Soundness:** 2 fair
**Presentation:** 2 fair
**Contribution:** 2 fair

**Summary:**

This paper proposes an implicit neural representation called Hybrid Sign and Distance Field (HSDF) for modeling surfaces with arbitrary topologies. It employs a neural network to regress unsigned and signed distance fields separately. It then fuses the two fields by taking the distance from the unsigned distance field and the sign of the signed distance field. In this way, the unsigned distance field allows modeling of open surfaces, and the signed distance provides the directions of the surfaces. They also simply modified the marching cube algorithm to a masked version to mesh their learned field.

The overall idea of this paper is simple and interesting. However, some motivations and arguments from the paper are problematic. The presentation quality of the paper could also be further improved.

**Questions:**

I am impressed by the numbers in Table 1. However, I have no idea what causes the difference? Could you please give more explanation about the time and memory? It would also be great to include IF-Net in this table as well.





**Limitations:**

The authors have mentioned some limitations in the text.

**Strengths And Weaknesses:**

Strengths:
1. I like the idea of modeling the surface by combing a signed distance field and an unsigned distance field, which enables arbitrary topology and surface direction modeling.
2. The idea of using two network heads to regress two fields is reasonable, as the combined one is not continuous.
3. According to Table1, the proposed method seems to be more time and memory efficient.



Weaknesses:
1. As stated in Line 30, the main motivation of the proposed method is that applying the marching cubes algorithm to UDF would convert all open surfaces into closed meshes. However, I don't think this point is valid, as we can post-process the generated meshes by Marching Cubes to represent open surfaces. Specifically, we can remove the faces in regions that do not contain the surface.
Alternatively, we can also apply the masked marching cubes to UDF to only extract regions containing the surfaces.
Moreover, NDF work has provided a method to output arbitrarily dense point clouds, which can then be used to estimate consistent normal direction accurately.
As a result, the point that "marching cubes algorithm cannot be applied to UDF for open surfaces" is very vague. I would highly suggest authors sell other benefits of learning a sign function and rewrite the paper in another way.

2. In section 3.4, the role of the signed function is not described. How does HSDF enable marching cubes? Why do we need a mask version of marching cubes? These points are not stated in the text.

3. Figure 5, for the NDF method, do you use the dense point cloud generation technique introduced in their paper? If points are dense enough, we will be able to estimate consistent normals, apply marching cubes, and flip the faces as a post processing for BPA.

4. In table 3, what's the meaning of only sampling 300 points. Much more points could be sampled to get a more accurate estimate.

5. In Table 2, it would also be great to include a direction-sensitive normal consistency (original version).

6. It would be better to include more recent UDF approaches in comparison.

7. The presentation could be improved a lot:

(a) Line 178: "closest surface normal direction" is confusing.

(b) Line 183: What does $\mathbb{R}_{0}$ mean?

(c) Line 184: $\mathbb{R}^{3} \mapsto[-1,1]$ should be {-1, 1} instead of an interval.

(d) Line 186: Please explicitly explain $\mathcal{S}_{\mathrm{x}}$.

(e) Line 192: For the signed distance, why is the clamping in one direction?

(f) Line 238: watertight meshes -> watertight meshes and open surfaces?

---

> ### Author Response · Authors · 2022-08-02
> **Response (2/2) to Reviewer oe6G**
>
> > Q5: In table 3, what's the meaning of only sampling 300 points. Much more points could be sampled to get a more accurate estimate.
>
> A5: We show the reconstruction evaluation from 300 points sampled mainly because:
> 1) We follow the same experiment setting as both NDF and IF-Net to ensure fair comparisons -- they also showed evaluations for sampling 300 points as input.
> 2) Evaluations in Table 3 focus on the watertight meshes without any complex inner structure, so the number of sampling points can be small.
>
> > Q6: In Table 2, it would also be great to include a direction-sensitive normal consistency (original version).
>
> A6: We have conducted the evaluations on the original version of normal consistency (NC) as shown below. All the results are reported on the test set with 3000 points as input. Due to the poor mesh surface quality reconstructed from dense point clouds using BPA, our method still outperforms NDF on the original NC metric.
>
> ||MGN|Car|Chair|Ship|Lamp|Mean
> |:-:|:-:|:-:|:-:|:-:|:-:|:-:|
> |NDF|0.938|0.817|0.865|0.850|0.869|0.868
> |Ours|**0.957**|**0.842**|**0.896**|**0.863**|**0.904**|**0.892**
>
> We will add this evaluation to the revision.
>
> > Q7: It would be better to include more recent UDF approaches in comparison.
>
> A7: There are concurrent papers that work on UDF representation, including MeshUDF [A] and Neural Dual Contouring [B]. We will provide comparisons with these approaches in the revision.
>
> [A] MeshUDF: Fast and Differentiable Meshing of Unsigned Distance Field Networks, arxiv paper.
>
> [B] Neural Dual Contouring, SIGGRAPH'22.
>
>
> > Q8: I am impressed by the numbers in Table 1. However, I have no idea what causes the difference? Could you please give more explanation about the time and memory? It would also be great to include IF-Net in this table as well.
>
> A8: We wish to clarify that the "Memory" in Table 1 actually refers to the memory cost of storing the output mesh. We apologize for the potential confusion caused and will clarify it in the revision. We provide the explanation about the time and memory below, followed by additional experiments on timing and memory cost with IF-Net evaluated.
>
>
> 1) Time: The ball pivoting algorithm (BPA) is a progressive method which picks a seed triangle and gradually grows the mesh using a rolling ball. Specifically, the ball connects all the vertices that it rolls over and stops when it falls off the boundary of the grown mesh. Such a method can hardly be accelerated in parallel as it would incur conflicts in the shared regions spanned by different seeds. In contrast, the marching cubes algorithm can interpolate triangles within each grid cell independently. Hence, it can be significantly accelerated using parallel computing, making it faster than the BPA approach.
>
> 2) Memory: The NDF method requires dense point cloud in order to produce reasonably good results. To convert the dense point cloud to mesh, the BPA algorithm simply connects all the vertices without removing a single point. In contrast, the marching cubes algorithm utilizes regular grids to sample field values and extract the triangles for each grid. Considering the sparsity of the surface occupancy, the vertices of the resulting mesh can be much fewer than the grid points themselves. Hence, our method requires much fewer memory for storing the output mesh compared to NDF.
>
>
> We also evaluate IF-Net on the same datasets including complex shapes. We report the quantitative results in the table below. IF-Net is faster than ours because it doesn't need to compute the masks for marching cubes. However, **IF-Net can't reconstruct open surfaces as we do**. In addition, IF-Net is prone to generate redundant artifact meshes when dealing with open surface, leading to larger mesh storage cost as shown in the table.
>
> Inference time (forward pass + meshing):
> ||$64^3$|$128^3$|$256^3$|
> |:-:|:-:|:-:|:-:|
> |NDF|89s|58m|780m|||
> |Ours|5s|17s|95s|||
> |IF-Net|<1s|3s|19s||
>
> Mesh storage consumption:
> ||$64^3$|$128^3$|$256^3$|
> |:-:|:-:|:-:|:-:|
> |NDF|17M|64M|1276M|||
> |Ours|1M|3M|10M|||
> |IF-Net|1.2M|5M|15M||
>
> We will include these results in the revision.
>
> ## Minor issues
>
> > Line 178: "closest surface normal direction" is confusing.
>
> We will modify it to "normal direction of its closest surface point".
>
> > Line 183: What does $R_0$ mean?
>
> It's a math notation for set {$ x|x∈R,x≠0 $}.
>
> > Line 186: Please explicitly explain $S_x$.
>
> We define S and X at lines 159 to 161. And $S_x$ stands for the target surface of a certain input point cloud X.
>
> > Line 192: For the signed distance, why is the clamping in one direction?
>
> Thanks for pointing this out! This is a typo which should be two-directional clamping. We will fix this typo in our revision.

---

> ### Author Response · Authors · 2022-08-02
> **Response (1/2) to Reviewer oe6G**
>
> We thank the reviewer for the insightful comments and suggestions! We provide our detailed response below.
> ## Main Questions
> > Q1: We can use Marching Cubes on UDF and post-process the generated meshes to represent open surfaces.
>
> A1: Since there are no negative values in UDF, we can only use a small positive level value (instead of zero in SDF) to extract meshes using the Marching Cubes algorithm. This typically leads to two layers of surfaces around the target surface (the zero-level surface). To reconstruct a open surface as simple as a 2D rectangle surface patch, simply applying marching cubes would generate a **closed mesh that tightly encloses the surface patch**. It remains extremely challenging to identify which faces do not contain the target surface or generate a mask to convert the closed mesh back to an open one by only using the UDF, not mention more complex shapes that contain both closed and open surfaces.
>
> In fact, the NDF [13] paper also recognizes the difficulty of using marching cubes to extract open surfaces from UDF, as we quoted from its introduction: *"Most classical methods, such as marching cubes and volume rendering, find the zero-level set by detecting flips from inside to outside and vice versa, which is **not possible with UDF**."* Hence, we believe we are working on a non-trivial problem that is well motivated.
>
> > Q2: The dense point clouds from NDF can be used to estimate consistent normal direction accurately.
>
> A2: First, the UDF 1) suffers from the vanishing gradient problem on the surface (also pointed out by [A]) and 2) relies on a heuristic process to iteratively push the points onto the surface using the gradient field. These limitations make UDF vulnerable to shapes with intricate geometry details: the point-pushing mechanism is easy to get stuck in the local minima (also pointed out by [B]). Hence, UDF may not be able to generate accurate dense point clouds for complex shapes in the first place.
>
> Second, even with an accurate dense point cloud, it remains an open question on how to estimate consistent and accurate normals for complex shapes with fine-grained details. This is due to the fact that dense point clouds don't include prior knowledge of face orientations in the dataset. This is manifested by our results in Figures 1, 6, and Table 2, where NDF fails to generate surfaces with consistent normal even if a very dense point cloud is used as input. In contrast, our sign field regression enables us to learn the correct face direction pattern from the training data. Therefore, our extracted meshes can have more consistent and accurate face normals compared to the NDF.
>
> [A] Deep Implicit Surface Point Prediction Networks, ICCV'21
>
> [B] 3PSDF: Three-Pole Signed Distance Function for Learning Surfaces with Arbitrary Topologies, CVPR'22
>
> > Q3: Why do we need a mask and how does it work on HSDF?
>
> A3: The signed field has been fused with the distance field as described in Section 3.3 to obtain our combined HSDF. As the signed field can estimate the inside/outside sign of query points for generating a zero-level set surface while the distance field can provide accurate point-to-surface distance, the fused HSDF can provide all necessary information for marching cubes to extract iso-surfaces.
>
> If the marching cubes is naively applied without a mask, it will generate a closed surface. Hence, if we want to generate open surfaces using the marching cubes algorithm, we need to provide a mask to stop the marching cubes procedure from producing meshes in the unwanted regions. This mask is computed from our distance field which enables us to perform marching cubes only in the regions that is close to our target surface. We will improve our statement in the revision.
>
> > Q4: Figure 5, for the NDF method, do you use the dense point cloud generation technique introduced in their paper? If points are dense enough, we will be able to estimate consistent normals, apply marching cubes, and flip the faces as post-processing for BPA.
>
> A4: As discussed in Section 1.4 of the supplemental material, we follow the exact same dense point cloud generation process in NDF and use the MeshLab scripts released by the author to apply the BPA and post-processing (closing holes and re-orienting faces). As mentioned in the response to Q2, due to the limitations of NDF itself and the difficulty of estimating accurate normals from unordered point clouds, the computed normal for BPA is not accurate enough. This leads to bad meshing quality for NDF. Further, as there are too many disconnected triangles and self-intersections in the outcome mesh, post-processing methods, including closing holes and face re-orienting, cannot work well. In contrast, our method can convert HSDF directly into a mesh using the marching cubes without generating a point cloud. Hence, we are able to generate results with much better meshing quality.

---

> > ### Comment · Reviewer_oe6G · 2022-08-09
> > **Thanks for the response!**
> >
> > Thanks for the response!
> >
> > According to my experience, after getting dense enough points, it's not hard to obtain high-quality meshes. In the NDF paper, they have also mentioned that "Since we are able to efficiently extract millions of points, naive classical algorithms for meshing [9] (which locally connect the dots) can be used to generate high-quality meshes."
> >
> > Would you mind sharing the generated dense point clouds (used in Fig 1,5,6) for me to have a try?

---

> > > ### Comment · Reviewer_nGbx · 2022-08-09
> > > **Meshing UDFs**
> > >
> > > According to [MeshUDF: Fast and Differentiable Meshing of Unsigned Distance Field Networks] the meshing procedure is indeed a huge burden. Despite the point cloud being arbitrarily dense with NDF's gradient descent, it is slightly noisy. Meshing it with the ball pivoting algorithm appears to be extremely slow and to yield rough meshes.
> > > Estimating normals and using marching cubes (as suggested by reviewer oe6G) would only work locally for open surfaces.
> > >
> > > This being said, I am also curious to see if an off-the-shelf point cloud meshing method can provide good results.

---

> > > > ### Author Response · Authors · 2022-08-09
> > > > **Thank you for the explanation**
> > > >
> > > > Thanks to the reviewer nGbx for further explanation, we provide some point cloud data[A] (mentioned by reviewer oe6G) and the reconstruction script[B] provided by NDF's author in these links below. If you are interested, you can try it for the reconstruction of the point cloud.
> > > >
> > > > [A] https://ufile.io/kdu0hoja
> > > >
> > > > [B] https://github.com/jchibane/ndf/issues/16

---

> > > ### Author Response · Authors · 2022-08-09
> > > **Response to Reviewer oe6G**
> > >
> > > Yes, it is a pleasure for us to share the generated dense point clouds (used in Fig 1,5,6) with you. The download link is provided in [A]. All the point clouds are generated following the official code released by NDF. The MeshLab meshing scripts are released by NDF's author in [B]. We use a BPA radius 0.01 (mentioned in sec 1.4 of supplemental material) instead of 0.005 in [B] since after lots of tests it demonstrates that the radius 0.005 would take a very long time to mesh even a single dense point cloud. In addition, in paper [C], they claimed: "In our experiments, we have found the ball-pivoting process to be very sensitive to this radius, and in many cases, it had to be tuned per-shape". We also found this in our experiments, so we would recommend experimenting with different radii of BPA for a more intuitive understanding of its limitations.
> > >
> > >
> > > [A] https://ufile.io/kdu0hoja
> > >
> > > [B] https://github.com/jchibane/ndf/issues/16
> > >
> > > [C] Deep Implicit Surface Point Prediction Networks (ICCV'21)

---

### Author Response · Authors · 2022-08-08
**A reminder for the discussion**

Dear AC and all reviewers:

Thanks again for all of your constructive suggestions, which have helped us improve the quality and clarity of the paper!

Since we have only one day left in the discussion phase, and we have not heard back from anyone yet regarding their post-rebuttal response. Please don’t hesitate to let us know if there are any additional clarifications or experiments that we can offer, as we would love to convince you of the merits of the paper. We appreciate your suggestions. Thanks!

---

### Meta-Review · Area_Chair_hnBu · 2022-08-27

**Recommendation:** Accept
**Confidence:** Less certain

**Metareview:**

The reviewers agree that the paper's idea to include both sign and distance fields is a valuable contribution to 3D computer vision research.

Reviewers ask sensible clarifying questions (e.g. orienting the training data, sign network continuity) and the rebuttal's answers are illuminating and to the point.

A short notice on terminoloy: I agree that "adversarial" should not be used here as it has a special meaning for the wider NeurIPS audience.  Regarding other wording suggestions, I add no extra vote for or against.


**Award:**

No

---

### Decision · Program_Chairs · 2022-09-14

Accept